# Tuning the Right Foundation Models is What you Need for Partial Label Learning

## Abstract

Partial label learning (PLL) seeks to train generalizable classifiers from datasets with *inexact* supervision, a common challenge in real-world applications. Existing studies have developed numerous approaches to progressively refine and recover ground-truth labels by training convolutional neural networks. However, limited attention has been given to foundation models that offer transferrable representations. In this work, we empirically conduct comprehensive evaluations of 11 foundation models across 13 PLL approaches on 8 benchmark datasets under 3 PLL scenarios. We further propose PartialCLIP, an efficient fine-tuning framework for foundation models in PLL. Our findings reveal that current PLL approaches tend to 1) achieve significant performance gains when using foundation models, 2) exhibit remarkably similar performance to each other, 3) maintain stable performance across varying ambiguity levels, while 4) are susceptible to foundation model selection and adaptation strategies. Additionally, we demonstrate the efficacy of text-embedding classifier initialization and effective candidate label filtering using zero-shot CLIP. Our experimental results and analysis underscore the limitations of current PLL approaches and provide valuable insights for developing more generalizable PLL models. The source code can be found in the supplementary material.

## 1 Introduction

Partial label learning (PLL) is an important weakly supervised learning framework and has been studied a lot in the past decade [1–8]. PLL aims to learn a classifier from datasets with *inexact* supervision in the label space, i.e., each training instance is associated with a set of candidate labels among which only one is correct. This framework alleviates the burden of precise data annotation, making it particularly valuable in scenarios where obtaining exact labels is costly or impractical. Therefore, PLL has been extensively studied across various real-world domains such as image annotation [9], web mining [10], ecoinformatics [11], and natural language processing [12].

The core challenge in PLL lies in accurately identifying the ground-truth label from the candidate label set. Existing PLL methods can be broadly categorized into two groups: average-based and identification-based methods. The average-based methods [13, 5] treat each candidate label equally by averaging the model outputs corresponding to all candidate labels. By contrast, identification-based methods [14, 15] progressively identify the ground truth label from the candidate label set through iterative refinement. Recently, deep neural network techniques have further enhanced PLL performance, such as PRODEN [16] and CRDPLL [17].

Despite these advancements, standard PLL (ST-PLL) methods often underperform in real-world scenarios, particularly in long-tailed PLL (LT-PLL) [18] and instance-dependent PLL (ID-PLL) [19] settings. This suboptimal performance can be attributed to the fact that ST-PLL assumes that the number of instances across all categories is uniform, and the false-positive labels in the candidate

Submitted to 39th Conference on Neural Information Processing Systems (NeurIPS 2025). Do not distribute.

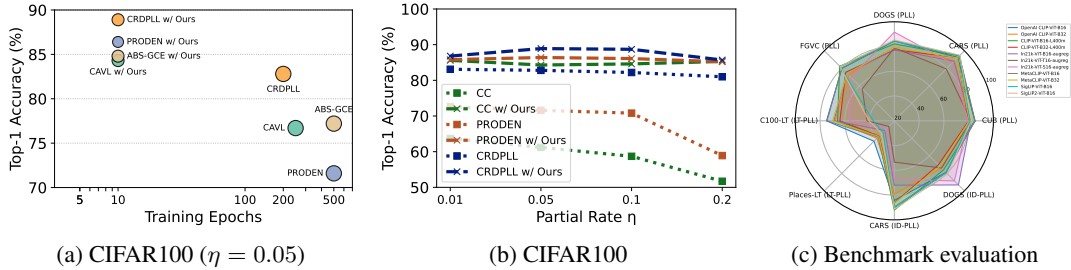

(a) CIFAR100 ($\eta = 0.05$)  (b) CIFAR100  (c) Benchmark evaluation

Figure 1: *(a)* Performance comparison between ST-PLL approaches and their PartialCLIP-enhanced variants in terms of test accuracy and the number of training epochs. Marker sizes represent the number of learnable parameters in each model. *(b)* The impact of partial rate on model accuracy across different ST-PLL methods. *(c)* Evaluation of various foundation models under three scenarios with CRDPLL (ST-PLL), RECORDS (LT-PLL), and POP (ID-PLL) serving as representative methods.

label sets are often generated randomly. Moreover, prevailing PLL methods predominantly employ convolutional networks such as ResNet [20]. Training these models typically requires 200–1,000 epochs to reach convergence, involving extensive parameter updates of the entire model. This process imposes substantial demands on computation and memory. Even with such investments, the quality of learned representations often degrades in highly ambiguous or imbalanced regimes, resulting in subpar classification performance.

In this work, we explore the efficacy of fine-tuning existing open-source foundation models on PLL benchmarks. Specifically, we empirically assess the classification performance across three PLL scenarios, i.e., ST-PLL, LT-PLL, and ID-PLL, with 11 models for 13 PLL approaches on 8 datasets. To facilitate comprehensive evaluations, we introduce PartialCLIP, a unified fine-tuning framework tailored to PLL. Experimental results demonstrate that PartialCLIP significantly enhances the performance of existing PLL approaches with markedly fewer training epochs (refer to Figure 1a). Furthermore, **our findings** reveal that current PLL approaches:

- exhibit remarkably similar performance to each other under the standard PLL and ID-PLL scenarios (see Figure 1b), indicating the effectiveness of transferred representations.
- maintain superior and stable performance across varying levels of label ambiguity, even under high ambiguity conditions (see Figure 1b).
- are susceptible to the choice of foundation models (see Figure 1c) and fine-tuning methods Table 5, underscoring the importance of selecting appropriate models.

Additionally, we explore the vision-language alignment capability of CLIP in two aspects: 1) classifier initialization and 2) candidate label filtering. **For classifier initialization,** we employ class-specific embeddings derived from CLIP's text encoder, prompted with ``a photo of a [CLASS]''. This approach leverages CLIP's semantic understanding to reduce reliance on extensive supervised data during training. **For candidate label filtering**, we filter out semantically irrelevant candidate labels according to the cosine similarities between image embeddings and textual prompts. Remarkably, pruning over $50\%$ of candidate labels does not degrade performance and even yields improvements. This finding shifts the emphasis from the quantity of candidate labels to the quality-driven selection.

In summary, **our contributions** are as follows. **1)** We propose an early foundation model benchmark for PLL, evaluated on eight datasets across three PLL settings; **2)** We present two ways of leveraging vision-language model alignment to address the challenges of learning from inexact labels; **3)** We identify three new findings of tuning foundation model in PLL to guide the future research.

## 2   Related Works

**Partial Label Learning** ST-PLL methods can be roughly divided into two categories, i.e., the averaged-based strategy (ABS) and the identification-based strategy (IBS). ABS excludes non-candidate labels and treats every candidate label equally. It averages the model output of all candidate labels for prediction [13, 5]. IBS views the ground truth label as a latent variable and gradually eliminates the label ambiguity during the training process [14, 15]. PRODEN [16] suggested a

strategy to progressively uncover the ground-truth label. CC [21] devised label disambiguation approaches that are provably risk-consistent and classifier-consistent from a mathematical perspective. LWS [22] introduced a suite of leveraged weighted loss functions. CAVL [23] utilizes the class activation value to guide the model in selecting the ground-truth label from the candidate label set during training. ABS-MAE and ABS-GCE [24] revisited the average-based strategy methods. Following MoCo [25], PiCO [26] incorporated the widely used contrastive loss into PLL. CRDPLL [17] applied consistency regularization [27] within the candidate label sets. PAPI [28] computed similarity scores between feature prototypes and instance embeddings. CROSEL [29] employed two models to sift out reliable samples from the dataset through cross-selection for the training stage.

**Long-Tailed Partial Label Learning** In contrast to ST-PLL, LT-PLL is more complex and challenging. Several works have begun to focus on LT-PLL in recent years. For instance, [30, 31]tackled it by employing over-sampling techniques and imposing regularization constraints. SoLar [18] regarded LT-PLL as an optimal transport problem and harnessed the Sinkhorn-Knopp algorithm [32] to obtain a rapid approximation. It confines the pseudo labels to adhere to the estimated class distribution priors. RECORDS [33] get insights from the perspective of logit adjustment [34]. It updates the global representations with momentum, thereby dynamically determining the class distribution. It cleverly combines with the existing ST-PLL methods and alleviates the model's bias towards head classes through dynamic logit adjustment. HTC [35] constructs two expert classifiers, each excelling in inferring head classes and tail classes separately.

**Instance-Dependent Partial Label Learning** In the above two PLL paradigms, the candidate set of each instance is randomly generated and has nothing to do with the instance itself. However, in the real-world, the labels that are prone to misclassification are typically highly similar to the ground-truth label. Therefore, a more practical ID-PLL was proposed. VALEN [36] was the first to introduce ID-PLL, featuring a two-stage disambiguation process. Stage one aimed to recover the latent label distribution of instances by an auxiliary model; stage two trained the model with the recovered distribution. ABLE [19] proposed an ambiguity-induced positive selection contrastive learning framework to disambiguate labels. POP [37] presented a method that progressively purifies the candidate label set and optimizes the classifier. IDGP [38] modeled the ID-PLL candidate label generation process, using categorical and Bernoulli distributions to simulate the ground-truth and noisy label generation, respectively. DIRK [39] proposed a self-distillation-based label disambiguation method with the student model trained under the guidance of the teacher model's output.

**Fine-Tuning Foundation Models** Recently, Transformer-based models like CLIP [40], which have been pre-trained on large-scale image-text data, have witnessed remarkable success. In image classification tasks, the performance of transformer-based models can be improved by fine-tuning. For example, CoOp [41]adopts learnable prompt vectors by minimizing prediction errors. Tip-Adapter [42], a training-free adaption method, directly configures the adapter using cache mode. LIFT [43] has proved theoretically and experimentally that a lightweight classifier, together with diverse PEFT strategies, can effectively address the long-tailed problem. The core of VPT [44] lies in the concept of visual prompts. By inputting carefully designed prompts into the model, it guides the model to learn more effective representation methods without changing the basic structure.

# 3 The Proposed PartialCLIP Framework

## 3.1 Preliminary

**Setting of Partial Label Learning** Let $\mathcal{X}$ represent the feature space, and let $\mathcal{Y}$ denote the *label space* with $K$ classes. Consider a training set $\mathcal{D} = \{(\boldsymbol{x}_i, S_i) \mid 1 \leq i \leq N\}$, where $\boldsymbol{x}_i \in \mathbb{R}^d$ is a $d$-dimensional feature vector and $S_i \subseteq \{0, 1\}^K$ represents the candidate label set corresponding to $\boldsymbol{x}_i$. Notably, $S_i$ contains the ground-truth label of $\boldsymbol{x}_i$ and false positives. The objective of the PLL is to learn a multi-class classifier $f : \mathcal{X} \rightarrow \mathcal{Y}$ from the training set $\mathcal{D}$. In this paper, we consider various generation strategies of candidate labels, including uniform sampling, flip probability sampling, and instance-dependent generation. We also consider the long-tailed distribution of candidate labels.

**Generation Strategies of Candidate Label Set** To systematically compare how candidate label sets are formed under different assumptions, we describe three prevalent generation strategies below:

- *Uniform Sampling Strategy (USS):* Given the true label $y_i \in \mathcal{Y}$, USS constructs each $S_i$ by choosing any subset of the remaining $K - 1$ labels with equal probability, yielding $2^{K-1}$ equally likely candidate sets.
- *Flip-Probability Sampling Strategy (FPS):* FPS includes each false-positive label independently with probability $\eta$. To ensure that $S_i \neq \{y_i\}$, if no label is flipped, one label is randomly selected and flipped.
- *Instance-Dependent Generation:* A lightweight neural network is employed to learn $\mathbf{x}_i \mapsto S_i$, producing candidate label sets whose false-positive labels depend on $\mathbf{x}_i$. The high inter-label similarity in candidate label sets increases disambiguation difficulty.

**Foundation Models** In this section, we use CLIP [40] as a representative of the foundation models. In experiments, we offer detailed results for different foundation models to demonstrate the effectiveness of the proposed fine-tuning framework. The CLIP model contains an image encoder $f_I$ and a text encoder $f_T$. The training of the CLIP model is based on contrastive learning, which aligns image and text features in a shared latent space. This training paradigm enables CLIP to perform zero-shot classification by aligning image and text representations effectively. The zero-shot inference process of the CLIP model for image classification is as follows. First, each class $j \in [K]$ is transformed into a sentence $\boldsymbol{l}_j$ using a template "a photo of a [CLASS$_j$]". Then, the text encoder $f_T$ processes $\boldsymbol{l}_j$ to a text feature $\boldsymbol{t}_j$, given by $f_T(\boldsymbol{l}_j)$. Given an input image $\boldsymbol{v}$, the image encoder outputs an embedding $\boldsymbol{i} = f_I(\boldsymbol{v})$. Finally, zero-shot classification is performed by computing the cosine similarity between $\boldsymbol{i}$ and each $\boldsymbol{t}_j$, and selecting the class with the highest score.

**Fine-Tuning Methods** Our fine-tuning framework is agnostic to different types of foundation models. In this work, we use the pre-trained CLIP as our default foundation model for its transferrable representations. We fine-tune CLIP using a partial label loss function, leveraging a parameter-efficient fine-tuning (PEFT) strategy to balance performance and computational cost. PEFT methods consistently outperform both full fine-tuning and linear probing. Details descriptions of the PEFT methods are provided in Appendix F, and their empirical comparisons appear in Section 4.

### 3.2 Techniques for Improving Partial Label Learning

**Classifier Initialization via Textual Embeddings** Notably, directly optimizing a randomly initialized classifier is found to have a negative impact on fine-tuning the model [43]. Therefore, it is crucial to set an appropriate initial state for the classifier. A straightforward method is to apply linear probing using re-weighted or logit adjustment loss. Another approach is to compute the class mean feature as initialization. However, these two approaches not only require extracting features of training data but also are not available with scarce tail-class data. To overcome it, we tend to leverage the semantic knowledge from the text modality of CLIP. For multi-model transformer-based models like CLIP, since its visual and text modality are interconnected, we can utilize the class names in the text modality to "activate" its visual task capabilities and thus initialize the classifier. Specifically, we use hand-crafted textual prompts (e.g., "a photo of a [CLASS]") and compute their features $\boldsymbol{t}_1, \cdots, \boldsymbol{t}_K$, which are then used to initialize the classifier weights $\boldsymbol{w}_1, \cdots, \boldsymbol{w}_K$. The above processes are completed before training.

**Effective Candidate Labels** The USS [21] and the FPS [16] strategy often introduce redundant false-positive labels since they are randomly generated. In this case, we refine and obtain *effective* candidate labels by consulting zero-shot CLIP confidences. Specifically, for the image $\boldsymbol{x}_i$, we compute a confidence vector $\boldsymbol{z}_i \in \mathbb{R}^K$ that represents the confidence of the image belonging to each class based on zero-shot CLIP. Then, we refine the initial candidate label set $S_i$ by selecting the class indices with the top-$k$ highest confidence scores, resulting in the refined set $\widehat{S}_i$ as follows:

$$\widehat{S}_i = S_i \bigcap \operatorname{argtop}_k(\boldsymbol{z}_i), \tag{1}$$

where $\operatorname{argtop}_k$ returns the indices of the $k$ highest confidences produced by the zero-shot CLIP. In practice, we can set $k = \frac{K}{2}$ for simplicity, and a smaller $k$ may increase the possibility that the ground-truth labels are erroneously removed from the candidate label set. Therefore, overly aggressive pruning may remove true labels when CLIP is less discriminative.

Beyond these techniques, PartialCLIP offers several compelling strengths:

- **Loss-Agnostic:** PartialCLIP is compatible with many existing partial label loss functions to fine-tune the pre-trained foundation models.

- **Model-Agnostic:** Although we use CLIP as the default model in our experiments, Partial-CLIP does not rely on specific types of foundation models and fine-tuning methods.

- **Efficient:** Based on the parameter-efficient fine-tuning, PartialCLIP only requires 10 epochs to achieve convergence in most datasets.

# 4 Empirical Evaluation of PartialCLIP

Our main goal is to investigate the effectiveness of fine-tuning foundation models across diverse PLL scenarios, datasets, algorithms, pre-trained models, and fine-tuning methods. To this end, we conduct experiments under 3 PLL scenarios, 8 datasets, 13 algorithms, 11 pre-trained foundation models, and 6 fine-tuning methods. First, we demonstrate the benefits of fine-tuning foundation models by not only showing their strong generalization performance but also robustness to different learning algorithms and partial rates. Next, we analyze how performance varies with different configurations of pre-trained model weights and PEFT methods.

**Baselines** For standard partial label learning (ST-PLL) algorithms, we implement seven baseline algorithms: CC [21], LWS [22], CAVL [23], PRODEN [16], CRDPLL [17], ABS-MAE [24], and ABS-GCE [24]. To address class imbalances in PLL, we consider three long-tailed partial label learning (LT-PLL) methods: Solar [18], RECORDS [33], and HTC [35]. Instance-dependent partial label learning (ID-PLL) encompasses algorithms such as ABLE [19], POP [37], and IDGP [38].

**Datasets** The datasets used for the experiments in ST-PLL are CIFAR10 [45] and CIFAR100 [45]. For LT-PLL, we conducts experiments on ImageNet-LT [46], Places-LT [46], CIFAR10-LT [47], and CIFAR100-LT [47]. ID-PLL encompasses CIFAR10 [36], CIFAR-100 [36], andfour fine-grained image datasets [48] [49], i.e., CUB-200-2011 (CUB200) [50], Stanford Cars (CARS196) [51], FGVC Aircraft (FGVC100) [52], and Stanford Dogs (DOGS120) [53]. All images are scaled to $224 \times 224$.

## 4.1 Advantages of Fine-tuning Foundation Models

### 4.1.1 Finding 1: Significant Performance Improvement

Experimental results demonstrate that fine-tuning foundation models achieves significant performance improvements compared with training a ResNet from scratch or using a pre-trained checkpoint. We conduct experiments in three scenarios, i.e., partial label learning with completely random, long-tailed, and instance-dependent candidate labels. In these experiments, we employ CLIP as the foundation model for its robust performance.

Table 1: Comparisons of different ST-PLL algorithms based on ResNet and CLIP on CIFAR-10 and CIFAR-100 datasets. **Bold** indicates superior results.

| Method | Backbone | CIFAR-10 | | | | CIFAR-100 | | | |
|---|---|---|---|---|---|---|---|---|---|
| | | $\eta = 0.1$ | $\eta = 0.3$ | $\eta = 0.5$ | $\eta = 0.7$ | $\eta = 0.01$ | $\eta = 0.05$ | $\eta = 0.1$ | $\eta = 0.2$ |
| CC | Wide-ResNet-34-10 | 88.8 | 86.7 | 83.8 | 77.6 | 63.7 | 61.2 | 58.7 | 51.7 |
| w/ PartialCLIP | CLIP-ViT-B/16 | **97.1** | **97.1** | **96.7** | **96.9** | **85.6** | **84.3** | **84.6** | **85.3** |
| LWS | Wide-ResNet-34-10 | 86.5 | 84.3 | 54.8 | **38.5** | 58.5 | 55.2 | 40.1 | **23.9** |
| w/ PartialCLIP | CLIP-ViT-B/16 | **96.8** | **96.8** | **96.7** | 14.5 | **82.5** | **80.9** | **59.0** | 14.8 |
| CAVL | Wide-ResNet-34-10 | 95.1 | 94.8 | 93.7 | 70.6 | 79.1 | 76.7 | 51.7 | 16.2 |
| w/ PartialCLIP | CLIP-ViT-B/16 | **97.0** | **97.1** | **97.0** | **97.0** | **85.8** | **85.3** | **85.4** | **84.9** |
| CRDPLL | Wide-ResNet-34-10 | **97.5** | **97.3** | **97.1** | 95.8 | 83.1 | 82.8 | 82.2 | 81.0 |
| w/ PartialCLIP | CLIP-ViT-B/16 | **97.5** | **97.3** | 96.8 | **96.3** | **86.8** | **88.9** | **88.7** | **85.7** |
| PRODEN | Wide-ResNet-34-10 | 91.2 | 91.1 | 89.8 | 86.5 | 72.6 | 71.6 | 70.8 | 58.9 |
| w/ PartialCLIP | CLIP-ViT-B/16 | **97.4** | **97.3** | **97.2** | **95.9** | **85.8** | **86.4** | **86.1** | **85.2** |
| ABS-MAE | Wide-ResNet-34-10 | 93.9 | 87.6 | 80.5 | 42.6 | 8.2 | 4.6 | 2.6 | 2.9 |
| w/ PartialCLIP | CLIP-ViT-B/16 | **96.9** | **96.8** | **96.9** | **96.6** | **85.1** | **83.9** | **84.4** | **84.0** |
| ABS-GCE | Wide-ResNet-34-10 | 94.7 | 93.5 | 90.0 | 78.8 | 79.1 | 77.2 | 34.4 | 13.1 |
| w/ PartialCLIP | CLIP-ViT-B/16 | **97.0** | **97.0** | **96.4** | **96.0** | **85.4** | **84.8** | **84.9** | **84.2** |

Table 2: Test accuracy for ID-PLL methods on CIFAR and fine-grained datasets. The backbone for vanilla algorithms on the CIFAR dataset is ResNet-34 trained from scratch, while the backbone for the fine-grained dataset is pre-trained on ImageNet. **Bold** indicates better results.

| Method | Backbone | CIFAR10 | CIFAR100 | FGVC100 | CUB200 | CARS196 | DOGS120 |
|---|---|---|---|---|---|---|---|
| Zero-shot | CLIP-ViT-B/16 | 87.2 | 64.4 | 23.1 | 55.5 | 62.5 | 61.9 |
| POP | ResNet-34 | 89.6 | 64.6 | **77.9** | 64.9 | **85.3** | 74.9 |
| w/ PartialCLIP | CLIP-ViT-B/16 | **97.2** | **82.6** | 73.0 | **70.1** | 84.3 | **78.9** |
| IDGP | ResNet-34 | 84.1 | 62.3 | **72.5** | 58.2 | 79.6 | 66.8 |
| w/ PartialCLIP | CLIP-ViT-B/16 | **97.1** | **82.4** | 61.4 | **62.2** | **83.8** | **78.1** |
| ABLE | ResNet-34 | 83.9 | 63.9 | **74.1** | 63.2 | **85.8** | 72.8 |
| w/ PartialCLIP | CLIP-ViT-B/16 | **97.3** | **82.1** | 73.5 | **70.9** | 84.4 | **80.4** |

**Results under the ST-PLL Scenario** Table 1 reports results on CIFAR-10 and CIFAR-100 datasets. Integrating existing ST-PLL methods into PartialCLIP consistently outperforms the baselines trained with a ResNet model from scratch. Figure 1a and Figure 1b further demonstrate these improvements. This is because the pre-trained transformer-based models have stronger classification capabilities. When the partial rate $\eta = 0.2$ on the CIFAR100 dataset, CAVL demonstrates an accuracy improvement of 68.7%, while ABS-MAE exhibits an enhancement reaching 81.1%. As observed from Table 1, when the partial rate is relatively high, the accuracy of the LWS algorithm is rather poor. The detailed analysis of the reasons and the corresponding solutions are attached in the appendix G.3

**Results under the ID-PLL Scenario** Table 2 presents ID-PLL results on common and fine-grained image datasets. We trained for 100 or 200 epochs to achieve convergence for different datasets. In ID-PLL, we utilize the ID-PLL false-positive label generation strategy proposed by VALEN [36] to produce instance-dependent false-positive labels. Specifically, the candidate label sets are generated based on the predicted probabilities of WideResNet. We select the top 10% of the labels predicted by WideResNet for each image into the candidate sets. This yields highly correlated candidate labels, making label disambiguation more difficult. We found that on the CIFAR datasets, integrating PartialCLIP led to a significant overall improvement. Particularly on the CIFAR-100 dataset, PartialCLIP achieves up to a 20.1% improvement, reflecting CLIP-ViT-B/16's strong

Table 3: Accuracy comparisons on CIFAR10-LT and CIFAR100-LT under various flipping probability $\eta$ and imbalance ratio $\gamma$. **Bold** indicates superior results.

| Method | CIFAR10-LT | | | | | | | |
|---|---|---|---|---|---|---|---|---|
| | $\eta = 0.3$ | | | | $\eta = 0.5$ | | | |
| | $\gamma = 100$ | $\gamma = 150$ | $\gamma = 200$ | $\gamma = 250$ | $\gamma = 100$ | $\gamma = 150$ | $\gamma = 200$ | $\gamma = 250$ |
| Solar | 79.5 | 74.6 | 70.7 | 68.2 | 75.7 | 70.2 | 64.3 | 60.6 |
| w/ PartialCLIP | **88.7** | **86.6** | **82.9** | **83.0** | **81.7** | **80.7** | **81.1** | **73.1** |
| RECORDS | 78.0 | 73.6 | 71.7 | 66.7 | 74.1 | 67.7 | 63.8 | 58.6 |
| w/ PartialCLIP | **92.8** | **91.0** | **88.6** | **86.0** | **91.2** | **86.5** | **83.3** | **81.6** |
| HTC | 85.7 | 82.5 | 80.6 | 78.1 | 83.4 | 79.8 | 77.7 | 72.4 |
| w/ PartialCLIP | **95.1** | **94.1** | **93.4** | **92.8** | **93.6** | **92.3** | **91.5** | **89.5** |
| Method | CIFAR100-LT | | | | | | | |
| | $\eta = 0.05$ | | | | $\eta = 0.1$ | | | |
| | $\gamma = 20$ | $\gamma = 50$ | $\gamma = 100$ | $\gamma = 150$ | $\gamma = 20$ | $\gamma = 50$ | $\gamma = 100$ | $\gamma = 150$ |
| Solar | 57.1 | 47.5 | 42.0 | 39.1 | 52.6 | 42.5 | 36.4 | 33.8 |
| w/ PartialCLIP | **79.9** | **75.7** | **71.8** | **68.9** | **78.8** | **73.3** | **69.2** | **63.9** |
| RECORDS | 57.6 | 49.0 | 43.4 | 39.8 | 54.7 | 45.5 | 40.5 | 37.4 |
| w/ PartialCLIP | **81.9** | **78.8** | **77.1** | **73.4** | **79.9** | **78.2** | **74.8** | **71.5** |
| HTC | 61.1 | 53.3 | 47.5 | 44.8 | 60.5 | 51.3 | 46.2 | 42.6 |
| w/ PartialCLIP | **77.1** | **72.6** | **68.5** | **59.8** | **73.8** | **67.4** | **63.0** | **60.6** |

representational power. On fine-grained tasks, CLIP's gains are mixed compared to ResNet-34 [20] pretrained on ImageNet, This may be caused by the following reasons: **1)** we initialize the classifier weights with class names in the text modality. On the common image classification datasets, class names like "dog" can effectively leverage CLIP's general knowledge. However, on fine-grained datasets, overly specialized and detailed class names, like those in the FGVC100 dataset (e.g., "747-300", "DC-6"), cannot "activate" CLIP because its pre-training tasks are usually not so fine-grained; and **2)** The ResNet-34 used for comparison was trained on ImageNet and fine-grained datasets used for evaluation are sampled from ImageNet.

**Results under the LT-PLL Scenario** Table 3 and Table 10 summarize LT-PLL performance using FPS to generate the candidate labels over 10 training epochs. Compared to the previous ST-PLL and ID-PLL scenarios, CLIP-ViT-B/16 yields larger gains over ResNet under the LT-PLL scenario. This is because LT-PLL faces the dual challenges of class imbalance and ambiguous labels, making it more difficult to obtain high-quality representations, especially for the tail classes. By leveraging the pre-trained CLIP model, the quality of the representations can be guaranteed. Specifically, on CIFAR10-LT, HTC [35] exhibits the best performance. However, RECORDS [33] leads on the more complex CIFAR100-LT, Places-LT, and ImageNet-LT datasets. Table 11 presents the accuracy comparison under different shots. As can be observed, RECORDS performs comparably to HTC and Solar [18] on head classes but significantly lags behind on middle and tail classes. HTC and Solar rely on the outputs of the model's classifier head to estimate the class distribution. In the early stages of the model, the outputs are biased towards the head classes, leading to inaccurate estimations of the class distribution, which in turn exacerbates the bias in the model's outputs. In contrast, RECORDS estimates the distribution through global representations. These global representations are obtained through pre-training and are relatively stable. Therefore, the estimated distribution is relatively accurate, which further balances the model's outputs.

### 4.1.2 Finding 2: Diminished Impact of Algorithm Choice

In addition, although there is a significant disparity when the method uses Wide-ResNet-34-10 [54] as the backbone (for example, on the CIFAR-100 dataset, when the partial rate $\eta$ is 0.2, CRDPLL [17] outperforms CC by 29.3%), the leading margin drops to 0.4% when CLIP-ViT-B/16 is employed as the backbone. This phenomenon mainly occurs under the scenarios of ST-PLL and ID-PLL, and it is not significant in the LT-PLL setting. During the fine-tuning process, the pre-trained representations remain unchanged. When Wide-ResNet-34-10 is used as the backbone, the quality of the model representations trained from scratch may vary considerably. Therefore, we infer that the results of partial label learning tasks are positively correlated with the quality of the learned representations.

### 4.1.3 Finding 3: Robustness to Varying Partial Rate

Furthermore, as can be seen in Figure 1b, our proposed method exhibits superior robustness to varying partial rates, maintaining stable performance with only marginal degradation even under conditions of increasing label ambiguity $\eta$ in the ST-PLL scenario. For example, when using CC [21] as the baseline and comparing different backbones, we can observe that when $\eta$ increases from 0.1 to 0.7, the accuracy of the model with Wide-ResNet-34-10 as the backbone drops by 11.1%. In contrast, the performance of the fine-tuned CLIP-ViT-B/16 only decreases by 0.2%. This is because ambiguous supervision information can severely disrupt the learning of representations. However, the pre-trained representations have a high quality and remain unchanged during the fine-tuning process.

### 4.2 Choosing the Right Foundation Model to Fine-Tune

**Impact of Foundation Models** Table 4 and Figure 1c present the comparison results of diverse backbones integrated within the PartialCLIP framework under three scenarios: standard PLL (CRD-PLL), long-tailed PLL (RECORDS), and instance-dependent PLL (POP). The results indicate that the optimal backbone varies depending on the specific scenario and dataset characteristics. Specifically, MetaCLIP [55] generally outperforms other backbone categories under the PLL scenario, while OpenAI CLIP [40] excels in the LT-PLL scenario. SigLip [56] demonstrates superior performance in fine-grained classification tasks across both the ST-PLL and ID-PLL scenarios. Overall, ViT [57] pretrained on ImageNet [58] generally lags behind the CLIP series in terms of performance. Specifically, In21k-ViT-T16-augreg is pretrained on ImageNet21k, while In21k-ViT-B16-augreg and In21k-ViT-S16-augreg are pretrained on ImageNet1k. On simple datasets such as CIFAR-10, the

Table 4: Performance of pre-trained models across 3 PLL settings. The imbalance ratio for CIFAR-100-LT is 100. The partial rates of DOGS and FGVC datasets in the ST-PLL setting are 0.01 and 0.01. For other datasets, their partial rates are consistent with those presented in Table 5.

| Pre-trained Model | ST-PLL | | | | | | LT-PLL | | ID-PLL | |
|---|---|---|---|---|---|---|---|---|---|---|
| | C10 | C100 | CUB | FGVC | CARS | DOGS | C100-LT | Places-LT | CARS | DOGS |
| OpenAI CLIP-ViT-B16 | 97.3 | 88.7 | 85.1 | 79.9 | 92.4 | 84.4 | 74.8 | **43.3** | 84.3 | 78.9 |
| OpenAI CLIP-ViT-B32 | 96.7 | 85.4 | 80.4 | 72.2 | 88.8 | 79.7 | 69.5 | 39.9 | 79.2 | 74.1 |
| CLIP-ViT-B16-L400m | 97.2 | 86.5 | 84.9 | **82.4** | 94.2 | 82.1 | 68.8 | 38.1 | **92.1** | 76.1 |
| CLIP-ViT-B32-L400m | 96.5 | 85.5 | 80.0 | 75.5 | 92.3 | 77.3 | 66.9 | 37.4 | 86.2 | 70.3 |
| In21k-ViT-B16-augreg | 97.4 | 86.9 | 81.8 | 74.1 | 88.0 | **93.2** | 73.2 | 31.0 | 72.0 | **93.2** |
| In21k-ViT-T16-augreg | 96.3 | 85.1 | 81.8 | 56.6 | 79.3 | 78.9 | 65.5 | 26.4 | 53.4 | 73.9 |
| In21k-ViT-S16-augreg | 96.7 | 84.8 | 79.6 | 69.7 | 85.8 | 91.6 | 68.7 | 28.4 | 67.4 | 88.6 |
| MetaCLIP-ViT-B16 | **98.6** | **90.1** | **85.2** | 80.1 | 94.3 | 84.6 | **77.3** | 38.5 | 89.5 | 78.8 |
| MetaCLIP-ViT-B32 | 98.0 | 89.1 | 80.6 | 74.1 | 92.0 | 79.3 | 74.6 | 37.8 | 86.5 | 72.0 |
| SigLIP-ViT-B16 | 97.1 | 86.1 | 81.2 | 82.1 | 94.5 | 84.5 | 65.6 | 33.0 | 90.4 | 79.4 |
| SigLIP2-ViT-B16 | 97.3 | 86.7 | 82.1 | 82.0 | **94.9** | 85.3 | 71.4 | 34.5 | 91.4 | 80.7 |

Table 5: Comparison of different parameter-efficient fine-tuning methods. The best results are highlighted in bold and the second-best results are underlined.

| Methods | ST-PLL | | | | LT-PLL | | ID-PLL | |
|---|---|---|---|---|---|---|---|---|
| | CIFAR10 | CIFAR100 | CUB200 | CARS196 | CIFAR100-LT | Places-LT | DOGS120 | FGVC100 |
| | $\eta = 0.3$ | $\eta = 0.1$ | $\eta = 0.01$ | $\eta = 0.01$ | $\eta = 0.1$ | $\eta = 0.05$ | $\eta = 0.1$ | $\eta = 0.1$ |
| Zero-Shot | 87.2 | 64.4 | 48.8 | 59.1 | 64.4 | 39.1 | 61.9 | 23.1 |
| Adaptformer | **97.1** | 84.6 | **85.7** | 92.2 | 74.8 | **43.3** | 78.9 | 73.0 |
| w/o text init | 97.1 | 85.1 | 84.4 | 92.2 | 34.8 | 32.3 | 79.4 | 64.1 |
| Adapter | 96.8 | 84.6 | 84.5 | 92.0 | 74.6 | 42.8 | 78.6 | 66.4 |
| VPT-Shallow | 96.1 | 82.2 | 80.8 | 87.9 | 70.2 | 42.3 | 77.9 | 54.8 |
| VPT-Deep | 97.0 | **85.2** | 84.6 | 92.0 | **75.7** | 42.9 | **79.0** | **74.1** |
| w/o text init | 97.1 | 84.8 | 84.1 | 92.0 | 68.0 | 34.9 | 79.4 | 72.6 |
| LoRA | 95.2 | 83.7 | 85.6 | **92.9** | 75.5 | 42.7 | 78.6 | 73.0 |
| BitFit | 96.0 | 84.4 | 77.3 | 91.9 | 74.5 | 41.8 | 78.3 | 71.4 |
| Linear probe | 93.6 | 74.8 | 78.0 | 83.6 | 53.5 | 37.6 | 65.9 | 41.5 |
| Full fine-tuning | 59.1 | 24.7 | 48.8 | 59.1 | 6.9 | 2.3 | 14.1 | 15.8 |

former's performance is comparable to the latter's. However, as datasets become more challenging, the latter outperforms the former significantly.

**Impact of Fine-Tuning Methods** PartialCLIP is a general framework that allows for the integration of various fine-tuning methods. In our experiments, we evaluate zero-shot CLIP, full fine-tuning, and six PEFT methods, i.e., *BitFit* [59], *VPT-shallow* [44], *VPT-deep* [44], *Adapter* [60], *LoRA* [61], and *AdaptFormer* [62] into PartialCLIP and compare their performance. Experiments are conducted across standard PLL (using CC for CIFAR-10 and CIFAR-100, and CRDPLL for fine-grained datasets), LT-PLL (RECORDS), and ID-PLL (POP) settings. As shown in Table 5, VPT-Deep and Adaptformer generally achieve the highest accuracy. In addition, the performance of PEFT outperforms that of the linear probe, indicating its remarkable advantages in optimizing model performance. Additionally, classifiers initialized with semantic text embeddings consistently surpass those with random initialization, indicating the importance of semantic-aware initialization in enhancing model performance across various settings.

## 4.3 Further Analyses

**Effect of Classifier Initialization** Table 5 demonstrates that initializing the classifier with class names significantly improves the performance. Given that CLIP is a large vision-language multi-modal model, the class names in the text modality serve as supervision information. The interconnectedness of textual and visual features activates the model's inherent general knowledge, leading to excellent

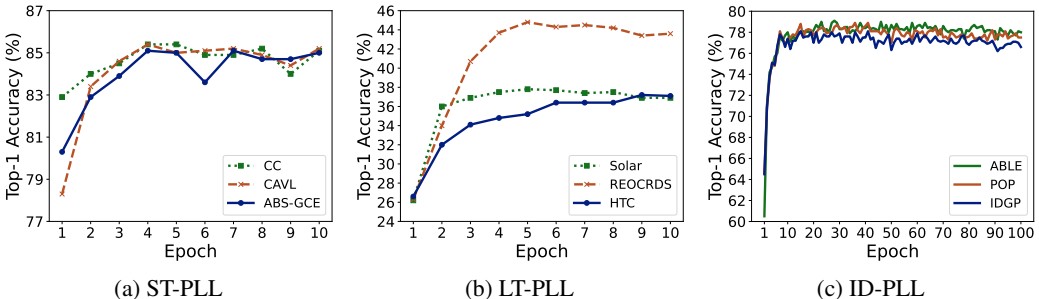

Figure 2: *(a)* Test accuracy curves of three ST-PLL methods on CIFAR100 where the partial rate $\eta$ is 0.1. *(b)* Test accuracy curves of three LT-PLL methods on Places-LT where the partial rate $\eta$ is 0.02. *(c)* Test accuracy curves of three ID-PLL methods on DOGS120.

Table 6: Effectiveness of the proposed CLIP candidate label filtering. For CIFAR datasets, we report the results of LWS method. For ImageNet-LT and Places-LT, the RECORDS algorithm is used.

|  | **CIFAR-10** | **CIFAR-100** | | **ImageNet-LT** | **Places-LT** |
|---|---|---|---|---|---|
| Settings | $\eta = 0.7$ | $\eta = 0.1$ | $\eta = 0.2$ | $\eta = 0.1$ | $\eta = 0.1$ |
| Zero-Shot CLIP | 87.2 | 64.4 | | 66.8 | 39.1 |
| Original Size | 7.3 | 10.9 | 20.8 | 100.9 | 37.4 |
| Effective Size | 3.8 | 5.9 | 10.8 | 50.9 | 19.1 |
| PartialCLIP | 14.5 | 59.0 | 14.8 | 58.1 | 37.8 |
| w/ CLIP pre-filter | **96.2** (+81.7) | **82.3** (+23.3) | **82.1** (+67.3) | **71.4** (+13.3) | **41.1** (+3.3) |

results in downstream image classification tasks. Figure 2 illustrates the test accuracy progression throughout the training process across three PLL scenarios.

**Impact of Candidate Labels Filtering** The experimental results in Table 6 demonstrate a strong correlation between the partial rate $\eta$ and the performance degradation of PartialCLIP. Specifically, as $\eta$ increases, a progressive decline in the performance metrics is observed. Notably, when the cardinality of the candidate label set exceeds a certain threshold, PartialCLIP's performance falls below that of direct zero-shot inference on the dataset. The efficacy of pre-filtering via CLIP stems from the observation that most candidate labels differ significantly from the ground-truth label. By leveraging the zero-shot ability of CLIP, a considerable number of false-positive labels can be excluded. This reduction in the candidate label set size mitigates interference during the disambiguation process, thereby enhancing overall model performance.

# 5   Conclusion

In this work, we propose PartialCLIP, a unified fine-tuning framework that leverages vision-language models for partial-label learning (PLL), including standard PLL, long-tailed PLL, and instance-dependent PLL. To the best of our knowledge, this is the first framework that systematically integrates vision-language models fine-tuning into these PLL scenarios. PartialCLIP incorporates 13 PLL baselines, 8 benchmark datasets, and 8 fine-tuning methods. Our experimental results demonstrate that PartialCLIP significantly outperforms previous convolutional network-based models, and pre-trained CLIP models exhibit robustness to label ambiguity and class imbalance, highlighting their potential for real-world weakly supervised learning scenarios. However, PartialCLIP relies heavily on the quality of pre-trained vision-language models, which may not capture fine-grained category distinctions. In the future, we aim to improve the fine-grained recognition capabilities of the framework by integrating advanced techniques and exploring more effective vision-language models.

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

## A    Implementation Details in PartialCLIP

For all experiments, we use the SGD optimizer with a batch size of 64, weight decay of $5 \times 10^{-4}$, and momentum of 0.9. For lightweight fine-tuning methods, the learning rate is 0.01. For full fine-tuning, we search the learning rate from {0.02, 0.01, 0.005, 0.002, 0.001, 0.0005}, considering its weak stability. In ST-PLL and LT-PLL tasks, our experiments indicate that convergence can be achieved in just 10 epochs. But in ID-PLL, because of the higher complexity of disambiguating candidate label sets, the model needs more training epochs. Specifically, on CIFAR-10, CIFAR-100, DOGS120, and CARS196 datasets, the model requires 100 epochs, and on FGVC100 and CUB200 datasets, it needs 200 epochs to converge. In PartialCLIP, we set the bottleneck dimension $r = 2^{\lfloor \log_2 (\frac{K}{2L}) \rfloor}$ for Adapter and AdaptFormer such that it learns even fewer parameters than the classifier (please refer to for detailed analysis). The scaling factor $\sigma$ of the cosine classifier is set to 25 (please refer to the corresponding paragraph for the analysis). All experiments are conducted on a single NVIDIA A6000 GPU. A GPU with 48GB of memory is sufficient for all reproductions. To meet different precision and storage needs, we provide three precision types: AMP [63], fp16, and fp32. fp16 saves space but has lower precision; fp32 offers higher precision with more storage consumption. AMP uses fp16 for memory storage to reduce memory usage and speed up data transfer, and switches to fp32 for critical operations like gradient updates, often with loss scaling to avoid gradient underflow. For data augmentation, we use RandAugment [64], Mixup [65], and CutMix [66].

## B    Details of Candidate Label Set Construction Strategies

**Uniform Sampling Strategy (USS) [21]:** In the USS strategy, it is assumed that the label space is of $K$ dimensions. In this case, apart from the ground truth label, each of the remaining $K - 1$ labels has two distinct states: either being included in the candidate label set or not being included. According to the principles of permutation and combination in combinatorial mathematics, the total count of all possible candidate label sets can be calculated as $2^{K-1}$. Moreover, regardless of the sizes, each possible candidate power label set has the same probability of occurrence.

**Flip Probability Sampling Strategy (FPS) [16]:** Within the FPS [16] strategy, a probabilistic approach is implemented for candidate label set construction. Specifically, for each instance, every false-positive label $\overline{y}$ can be incorporated into candidate label sets with a fixed probability parameter $\eta$. To ensure the integrity of the learning framework, a safeguard mechanism is implemented. When the random sampling process results in zero label flips for a particular instance, the system automatically selects and inverts one false label through a uniform random selection process, thereby guaranteeing at least one label modification per instance.

**Instance-Dependent Generation [36]:** Existing studies in ST-PLL and LT-PLL typically assume that each false label has a random or fixed probability of being included in the set of candidate labels. However, in practice, annotators tend to select candidate labels that are semantically related to the true label, resulting in instance-dependent candidate labels. It uses a lightweight neural network to generate instance-specific candidate label sets tailored to the characteristics of each sample. The candidate labels within these sets exhibit high similarity, thereby increasing the complexity of the disambiguation process.

## C    Core Components of PartialCLIP

As illustrated in Figure 3, the code architecture of PartialCLIP is meticulously organized into four distinct components: Config, Algorithm, Models, and Trainer.

**Configuration module**: The configuration layer systematically enumerates essential parameters required for PartialCLIP implementation, comprising two principal components: (1) Data Configuration, specifying dataset-related parameters including dataset nomenclature and storage path; (2) Model Configuration, governing training protocol specifications such as backbone type, fine-tuning paradigm, batch size, and gradient descent optimization rate.

**Algorithm module**: The algorithm layer incorporates three distinct partial label learning paradigms under the PartialCLIP framework: ST-PLL, LT-PLL and ID-PLL. This taxonomy systematically organizes state-of-the-art methodologies. ST-PLL Implements seven baseline algorithms: CC [21], LWS [22], CAVL [23], PRODEN [16], PiCO [26], CRDPLL [17], ABS-MAE [24], ABS-GCE [24].

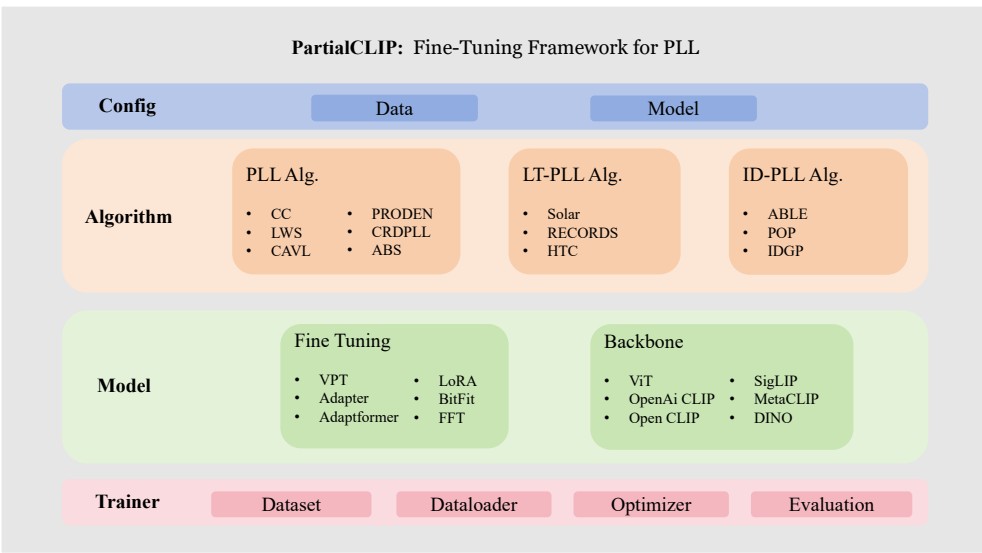

Figure 3: Code structure of PartialCLIP.

LT-PLL addresses long-tailed distribution scenarios through: Solar [18], RECORDS [33], and HTC [35]. ID-PLL includes ABLE [19], POP [37], and IDGP [38].

**Model module**: The model layer constitutes the core computational infrastructure of the PartialCLIP framework, comprising two principal components: Fine-Tuning and Backbone. The Fine-Tuning module support methods include VPT [44], Adapter [60], Adaptformer [62], LoRA [61], BitFit [59], Linear Probe and FFT. The Backbone module integrates multi-modal architectures like ViT [57], OpenAi CLIP [40], MetaCLIP [55], SigLIP [56], Open CLIP [67].

**Trainer module**: The trainer layer is responsible for the entire training process of the model. The Dataset part pertains to the construction of the dataset. The Dataloader is primarily utilized to load the dataset. The optimizer represents the optimizer module, and Evaluation is in charge of assessing the performance of the model.

# D    Statistics of Datasets

Table 7: Details of the built-in dataset of PartialCLIP, including the dimensions of data samples, the number of training data, the number of test data, and the number of categories.

| Dataset | # Dimensions | # Training data | # Test data | # Class |
|---------|--------------|-----------------|-------------|---------|
| CIFAR-10 | 224×224 | 50,000 | 10,000 | 10 |
| PLCIFAR10 | 224×224 | 50,000 | 10,000 | 10 |
| CIFAR-100 | 224×224 | 50,000 | 10,000 | 100 |
| Places-LT | 224×224 | 62,500 | 10,000 | 365 |
| ImageNet-LT | 224×224 | 115,800 | 10,000 | 1,000 |
| FGVC Aircraft (FGVC100) | 224×224 | 6,776 | 3,333 | 100 |
| Stanford Dogs (DOGS120) | 224×224 | 12,000 | 8,580 | 120 |
| Stanford Cars (CARS196) | 224×224 | 8,144 | 8,041 | 196 |
| CUB-200-2011 (CUB200) | 224×224 | 5,994 | 5,794 | 200 |

**CIFAR-10**

Table 8: Details of three versions of CIFAR-10 and CIFAR100 in PartialCLIP. $\gamma$ indicates the imbalance ratio in LT-PLL.

| Setting | Dataset | # Maximum class | # Minimum class | # Test data | # Class |
|---|---|---|---|---|---|
| ST-PLL | CIFAR-10 | 5,000 | 5,000 | 10,000 | 10 |
| ID-PLL | CIFAR-10 | 5,000 | 5,000 | 10,000 | 10 |
| LT-PLL | CIFAR-10-LT | 5,000 | $\lfloor \frac{5,000}{\gamma} \rfloor$ | 10,000 | 10 |
| Real-World PLL | PLCIFAR10 | 5,000 | 5,000 | 10,000 | 10 |
| ST-PLL | CIFAR-100 | 500 | 500 | 10,000 | 100 |
| ID-PLL | CIFAR-100 | 500 | 500 | 10,000 | 100 |
| LT-PLL | CIFAR-100-LT | 500 | $\lfloor \frac{500}{\gamma} \rfloor$ | 10,000 | 100 |

The CIFAR-10 [45] dataset is a natural image ($32\times32$ pixels) recognition dataset consisting of 10 classes. There are 50000 training samples and 1000 test samples per class. Considering the patch size of CLIP-ViT-B/16 is 16, we resize the CIFAR-10 dataset to $224\times224$.

**CIFAR-100**

The CIFAR-100 [45] dataset is a natural image ($32\times32$ pixels) recognition dataset consisting of 100 classes. There are 500 training samples and 100 test samples per class. We also resize the CIFAR-100 dataset to $224\times224$.

**Places-LT** The Places-LT [46] dataset is of great importance in computer vision, containing 62,500 images from 365 classes, with the number of images per class ranging from 5 to 4,980. It is mainly applied to scene recognition tasks in computer vision, enabling researchers to train and evaluate related models.

**ImageNet-LT** ImageNet-LT [46] is a significant dataset in the field of computer vision. It consists of 115,800 images distributed across 1000 classes. The number of images per class varies greatly, with a maximum of 1280 images for some classes and a minimum of only 5 images for others. This large variance in class sizes poses unique challenges for machine learning algorithms, especially in terms of handling class imbalance.

**Stanford DOGS120** Stanford DOGS120 [53] focuses on the recognition of dog images. It contains a total of 120 different dog breeds, with each breed having a varying number of sample images ranging from 150 to 200. The images in this dataset typically have a resolution of around $224\times224$ pixels. For each class, 100 images are allocated for training, while the remaining images (at least 50 per class) are reserved for testing.

**Stanford Cars196** Stanford Cars196 [51] is a dataset used for car image recognition. It consists of images of 196 different classes of cars. The number of images within each class ranges from approximately 30 to 100, offering a diverse set of samples for each car class. The images in this dataset generally have a resolution of $224\times224$ pixels. The training set consists of a total of 8,144 images, and the test set contains 8,041 images.

**FGVC Aircraft** FGVC Aircraft (FGVC100) [52] is mainly designed for fine-grained visual categorization tasks. It contains 100 different fine-grained categories. There are a total of 10,000 images. The number of images in the training set is 6,667, and the remaining ones constitute the test set. The images in the FGVC100 dataset generally have a resolution of $224\times224$ pixels.

**CUB-200-2011** CUB-200-2011 [50], also known as the Caltech-UCSD Birds-200-2011 dataset, is centered around bird image recognition. It consists of 200 different species of birds. The total number of images is 11,788. The size of the training set is 5,994, and the size of the test set is 5,794. The images in this dataset usually have a resolution of $224\times224$ pixels.

# E Details of Implemented PLL algorithms in PartialCLIP

## E.1 ST-PLL baselines

**PRODEN [16]** is designed to approximately minimize the proposed risk estimator by relaxing the minimization problem into a weighted combination. This approach integrates the learning of weights and the classifier in a unified manner, effectively mitigating the risk of overfitting.

**RC and CC [21]** devised label disambiguation approaches that are provably risk-consistent and classifier-consistent.

**LWS [22]** introduced a family of loss functions for partial label learning, termed the Leveraged Weighted (LW) loss, which incorporates a leverage parameter $\beta$ to balance the trade-offs between losses on partial labels and non-partial labels.

**CRDPLL [17]** applied consistency regularization [27] within the candidate label sets. Meanwhile, it derives entirely accurate supervision from the non-candidate labels, ensuring that the complements of the candidate labels are unequivocally excluded from being the ground-truth labels.

**CAVL [23]** leverages the Class Activation Value (CAV), and it guides the model to pick the ground truth label from candidates during training. It turns PLL into supervised learning, enabling the model to recognize true labels using learned CAV-based representation.

**ABS-MAE and ABS-GCE [24]** refocused on the average-based strategy (ABS) methods. Theoretically, it introduced five data generation processes for noise-free and noisy partial labels, thereby addressing a critical gap in the theoretical understanding of PLL robustness. Empirically, it conducted comprehensive experiments to validate its theoretical insights.

## E.2 LT-PLL baselines

**SoLar [18]** conceptualizes the LT-PLL as an optimal transport problem, leveraging the Sinkhorn-Knopp algorithm [32] to achieve an efficient approximation. This approach ensures that the generated pseudo-labels conform to the estimated class distribution priors.

**RECORDS [33]** adopts a logit adjustment perspective [34], dynamically updating global representations through momentum to infer the class distribution. By integrating with existing partial label learning methodologies, it mitigates model bias towards head classes via dynamic logit adjustment.

**HTC [35]** employs a dual-expert classifier framework, where each classifier specializes in the inference of head and tail classes, respectively. It incorporates a classifier weight estimation (CWE) module, designed to discern the class affiliation of a sample—whether it pertains to a head class or a tail class. This module adaptively adjusts and fuses the outputs from the dual classifiers, thereby enhancing the accuracy of the final prediction.

## E.3 ID-PLL baselines

**ABLE [19]** introduced an ambiguity-induced positive selection contrastive learning framework aimed at resolving label ambiguity. It jointly optimizes a representor that minimizes a weighted sum of contrastive losses across all groups and a classifier that minimizes a classification loss.

**POP [37]** progressively refined the learning model and purified the candidate label sets in each training epoch. Theoretically, POP expands reliable model regions efficiently. Technically, POP is compatible with arbitrary PLL losses and improves their performance in instance-dependent cases.

**IDGP [38]** formulated the candidate label generation process in ID-PLL, employing categorical and Bernoulli distributions to model the generation of ground truth labels and false-positive labels, respectively.

# F    Details of PEFT Methods

## F.1    Adapter

Adapter [60] is a technique in machine learning and deep learning. Typically, an adapter works by adding a small set of trainable parameters to the pre-trained model. These parameters are trained to capture the specific characteristics of the new task or domain, while keeping the majority of the original model's parameters fixed. This enables efficient transfer learning, reducing data and resource requirements for fine-tuning, and is effective in various natural language processing tasks.

## F.2    Adaptformer

AdaptFormer [62] replaces the MLP block in the Transformer encoder with AdaptMLP. AdaptMLP consists of two parallel sub-branches. The left-hand branch contains an MLP layer identical to that in the original network, termed the frozen branch. The right-hand branch is a newly introduced task-specific lightweight module, designed as a bottleneck structure. This lightweight encoder-decoder architecture aims to limit the number of newly added parameters by reducing the intermediate dimension. In practice, this design has demonstrated remarkable efficacy.

## F.3    LoRA

LoRA [61], short for Low-Rank Adaptation, is a technique used to fine-tune transformer-based models. It freezes the pre-trained parameters of the original model and only adapts a small number of newly added low-rank matrices. This significantly reduces the storage and computing resources required for fine-tuning, making it more efficient and cost-effective. At the same time, LoRA can achieve similar performance to traditional fine-tuning methods. It has been widely used in various natural language processing tasks and has become an important method in the field of large language model optimization.

## F.4    BitFit

BitFit [59] is a method in the field of machine learning, particularly for fine-tuning pre-trained transformer-based models. It focuses on adapting the bias terms of the model while keeping the other parameters fixed. By doing so, it aims to achieve efficient adaptation to new tasks with minimal computational cost and without significantly altering the pre-learned knowledge of the model.

## F.5    VPT

Visual prompt tuning [44] is a technique in the field of computer vision. It aims to adapt pre-trained models to specific tasks by adding and tuning visual prompts. These prompts can be in the form of image-based cues. This method enables more efficient fine-tuning with fewer parameter adjustments, enhancing the model's performance on targeted visual tasks.

# G    Additional Experimental Results

## G.1    Results on Real-world Dataset PLCIFAR10

In addition to the previous three simulated PLL settings, we also test the performance of PartialCLIP on a real-world PLL dataset PLCIFAR10 [68], which is created through manual annotation and is divided into two types: PLCIFAR10-Aggregate and PLCIFAR10-Vaguest. PLENCH also proposed two evaluation metrics on the validation set: covering rate (CR) and oracle accuracy (OA). We used CLIP-ViT-B/16 as the backbone in PartialCLIP and compared the results with those in PLENCH that used ResNet as the backbone. For all baselines combined with PartialCLIP, we set the number of training epochs to 10. According to Table 9, we found that almost all metrics of each baseline have been improved to varying degrees.

Table 9: Classification Accuracies of PLL methods on PLCIFAR10 dataset. **Bold** indicates better results.

| Methods | Backbone | Aggregate | | Vaguest | |
|---|---|---|---|---|---|
| | | w/ CR | w/ OA | w/ CR | w/ OA |
| CC | ResNet | 80.7 | 81.4 | 71.8 | 70.1 |
| w/ PartialCLIP | CLIP-ViT-B/16 | **95.7** | **96.2** | **88.5** | **93.6** |
| LWS | ResNet | 55.3 | 55.5 | 60.2 | 61.0 |
| w/ PartialCLIP | CLIP-ViT-B/16 | **95.9** | **95.9** | **94.9** | **95.1** |
| CAVL | ResNet | 68.1 | 68.2 | 63.6 | 63.7 |
| w/ PartialCLIP | CLIP-ViT-B/16 | **77.1** | **77.7** | **79.6** | **80.4** |
| CRDPLL | ResNet | 81.6 | 81.7 | 76.2 | 75.7 |
| w/ PartialCLIP | CLIP-ViT-B/16 | **87.7** | **87.6** | **90.0** | **91.9** |
| PRODEN | ResNet | 86.0 | 85.9 | 75.0 | 74.8 |
| w/ PartialCLIP | CLIP-ViT-B/16 | **95.0** | **95.7** | 62.9 | 68.8 |
| ABLE | ResNet | 85.9 | 86.1 | 75.5 | 74.9 |
| w/ PartialCLIP | CLIP-ViT-B/16 | **93.0** | **95.6** | **92.4** | **92.7** |
| POP | ResNet | 85.0 | 85.0 | 75.2 | 74.3 |
| w/ PartialCLIP | CLIP-ViT-B/16 | **95.0** | **95.2** | **93.2** | **93.8** |
| IDGP | ResNet | 82.8 | 83.4 | 76.1 | 76.1 |
| w/ PartialCLIP | CLIP-ViT-B/16 | **96.2** | **96.3** | **94.1** | **94.1** |

Table 10: Test accuracy of different LT-PLL methods with PartialCLIP on Places-LT and ImageNet-LT under various flipping probability $\eta$. **Bold** indicates superior results.

| Methods | Places-LT | | | | ImageNet-LT | | | |
|---|---|---|---|---|---|---|---|---|
| | $\eta = 0.01$ | $\eta = 0.02$ | $\eta = 0.05$ | $\eta = 0.1$ | $\eta = 0.01$ | $\eta = 0.02$ | $\eta = 0.05$ | $\eta = 0.1$ |
| Solar | 37.7 | 37.8 | 36.1 | 32.4 | 60.4 | 60.0 | 55.5 | 47.9 |
| RECORDS | **45.4** | **44.8** | **43.3** | **37.8** | **72.7** | **73.1** | **70.7** | **58.1** |
| HTC | 40.2 | 37.2 | 31.3 | 21.2 | 57.9 | 49.5 | 36.4 | 25.0 |

Table 11: Different shots accuracy comparisons on Places-LT ($\eta = 0.05$) and ImageNet-LT ($\eta = 0.01$). The best results are marked in bold, and the second-best are marked underlined.

| Methods | Places-LT | | | ImageNet-LT | | |
|---|---|---|---|---|---|---|
| | Many | Medium | Few | Many | Medium | Few |
| Solar | 52.8 | 34.2 | 9.5 | **82.4** | 55.7 | 14.6 |
| RECORDS | **55.3** | **42.0** | **24.0** | 78.7 | **73.7** | **52.7** |
| HTC | 53.2 | 25.1 | 5.4 | 82.2 | 52.1 | 9.4 |

## G.2 More Results on LT-PLL datasets

## G.3 Investigation into the Deterioration of the LWS Algorithm's Performance and Tailored Countermeasures

We observed that when Wide-ResNet-34-10 is used as the backbone, as the partial rate increases from 0.1 to 0.7, the accuracy of LWS [22] on the CIFAR100 dataset drops from 86.5% to 38.5%, with a decline rate reaching 48%. Meanwhile, when CLIP-ViT-B/16 is employed as the backbone, the accuracy can remain relatively stable when the partial rate is relatively low. However, when the partial rate increases to 0.7, a "collapse" phenomenon also occurs in performance.

To explore the reasons behind the observed performance changes, it is essential to delve into the principles of the LWS algorithm. In LWS, the leverage parameter $\beta$ is incorporated into the loss functions. This parameter serves to trade off the losses associated with partial labels and those of non-partial labels. Specifically, the partial loss function under consideration assumes the form.

$$\bar{\mathcal{L}}_\psi(\vec{y}, g(x)) = \sum_{z \in \vec{y}} w_z \psi(g_z(x)) + \beta \cdot \sum_{z \notin \vec{y}} w_z \psi(-g_z(x)),$$

where $\vec{y} \in \vec{\mathcal{Y}}$ denotes the partial label set. It consists of a binary loss function $\psi(\cdot) : \psi(x) = \frac{1}{1+e^x}$, weighting parameters $w_z \geq 0$ on $\psi(g_z)$ for $z \in \mathcal{Y}$, and the leverage parameter $\beta \geq 0$ that distinguishes between partial labels and non-partial ones.

However, LWS focuses on differentiating between candidate labels and non-candidate labels, making their boundaries clear. This leads to the following situation: when $\eta$ is small, the candidate label set is relatively small, and most non-candidate labels can be excluded. However, when $\eta$ increases, the candidate label set also expands. The excluded non-candidate labels only account for a small proportion, and the size of the sample space of candidate labels even approaches that of the entire sample space. This results in very weak supervision information provided, and the difficulty is close to that of an unsupervised task. Moreover, within the candidate set, LWS only uses a simple binary classification loss, without additional designs like those in CRDPLL. This makes it extremely difficult to identify the ground truth label within the candidate set.

Therefore, considering that LWS is highly sensitive to the size of the candidate label set, we utilize the zero-shot capability of CLIP to pre-filter the candidate label set before training, excluding those false-positive labels that can be distinguished by general knowledge alone. In the case of CIFAR-100 dataset with $\eta = 0.2$, since the candidate set is relatively large, we select the top 30% of the labels based on the results of CLIP zero-shot. In other cases, we select the top 50% of the labels in terms of confidence for each sample. We found that after pre-screening the candidate label set and then conducting the training, the performance is significantly improved, comparable to the results obtained under low partial rates. Specifically, in the context of the CIFAR10 dataset with a partial rate of 0.7, the test accuracy was increased from $14.5\%$ to $96.2\%$. For the CIFAR100 dataset, when the partial rates were 0.05, 0.1, and 0.2 respectively, the test accuracies were increased from $80.9\%$, $59.0\%$, and $14.8\%$ to $81.9\%$, $82.3\%$, and $82.1\%$ respectively.

## H   Limitations and Broader Impacts

**Limitations** Although PartialCLIP integrates a certain number of PLL baselines, there are still some methods and frameworks that are incompatible with it. How to equip our framework with more algorithms is a question worthy of further exploration.

**Broader Impacts** This research falls within the field of weakly supervised learning, which aims to optimize performance while reducing data labeling costs. As its effectiveness is increasingly validated and applications grow, reliance on comprehensive data annotation may decline. This could potentially lead to higher unemployment rates among data annotation professionals, underscoring the need for proactive measures to address associated socioeconomic impacts.

