# OpenReview forum: "Tuning the Right Foundation Models is What you Need for Partial Label Learning"
_NeurIPS.cc/2025/Datasets_and_Benchmarks_Track — Submitted to NeurIPS 2025 Datasets and Benchmarks Track_

### Official Review · Reviewer_icVt · 2025-06-01

**Rating:** 5
**Confidence:** 4

**Summary:**

This paper explores the application of large foundation models, such as CLIP, to enhance partial label learning (PLL)—a weakly supervised learning paradigm where each training instance is associated with multiple candidate labels, but only one is correct. The authors evaluate 11 foundation models across 13 PLL methods and several datasets on three PLL scenarios (standard, long-tailed, and instance-dependent), demonstrating that these models substantially improve PLL performance and often yield consistent results across methods. They propose PartialCLIP, an efficient fine-tuning framework that leverages foundation models, highlighting the critical role of model selection and adaptation for achieving optimal performance. Overall, the study underscores the importance of properly tuning foundation models to improve label disambiguation and classification in PLL tasks.

**Additional Feedback:**

Please see the limitations and weaknesses.

**Dataset Code Accessibility:**

Yes

**Ethical Considerations:**

No, there are no or only very minor ethics concerns

**Limitations Weaknesses:**

1. It seems CLIP is very effective to generate reliable confidence scores, it is better to give the experiment results with the following updated candidate label set $\widehat{S}_i$:

$\widehat{S}_i = S_i \cap \operatorname*{argtop}(\mathbf{z}_i)$

2. The top-k selection is important for the proposed method. It is better to give the sensitivity analysis about the model to different k values.

3. The paper lacks of explanation on the generation process of  partial label and instance-dependent partial label.

**Strengths Contributions:**

1. The authors claims that it is the first attempt to integrate vision-language models fine-tuning into the PLL scenarios. The proposed method innovatively integrates CLIP's zero-shot capabilities for improving the PLL.

2. The paper is well written and easy to follow. Furthermore, the code has been made public to enhance its reproducibility.

3. The extensive experiments underscore the effectiveness of the proposed method by perform experiments with 3 PLL scenarios, 8 datasets, and 11 models.

---

> ### Author Rebuttal · Authors · 2025-07-29
>
> Thank you for your time in reviewing our work; your insights have provided important inspiration. Presented next is a compendium of your comments coupled with our appropriate reactions.
>
> ### Q1 & Q2: It seems CLIP is very effective to generate reliable confidence scores, it is better to give the experiment results with the following updated candidate label set
>
> **A1&A2:**  Below are sensitivity experiments on the top-k accuracy of CLIP's zero-shot results, along with our model's corresponding performance across the four datasets referenced in Table 6 of the paper. Specifically, for each sample's CLIP zero-shot inference output, we analyze the proportion of samples where the ground-truth label is included in the top k classes—with k accounting for 10% to 80% of the total class count K. This is exactly what "Top-k Accuracy" refers to in the first table's title below. Results show that when k = 50%*K, all four datasets guarantee that at least 98.7% of samples keep their ground-truth labels.
>
>
> Table: Top-k Accuracy of CLIP Zero-Shot Inference under Different k
>
> |        |   CIFAR10   |   CIFAR100   |    Places-LT   |    ImageNet-LT   |
> | :-----: | :---------: | :----------: | :------------: | :--------------: |
> |  10%   |    90.15    |     91.98    |      93.04     |       99.44      |
> |  20%   |    96.18    |     95.54    |      96.99     |       99.76      |
> |  30%   |    97.82    |     97.11    |      98.40     |       99.87      |
> |  40%   |    98.69    |     98.08    |      99.13     |       99.93      |
> |  50%   |    99.32    |     98.72    |      99.51     |       99.95      |
> |  60%   |    99.72    |     99.22    |      99.72     |       99.97      |
> |  70%   |    99.86    |     99.53    |      99.84     |       99.99      |
> |  80%   |    99.94    |     99.75    |      99.93     |       99.99      |
>
>
> Table: Model Performance under Different k
>
> |        |   CIFAR10   |   CIFAR100   |    Places-LT   |    ImageNet-LT   |
> | :-----: | :---------: | :----------: | :------------: | :--------------: |
> |  10%   |    92.2     |     80.2     |      34.3      |       71.0       |
> |  20%   |    93.7     |     81.3     |      39.3      |       68.7       |
> |  30%   |    94.6     |     81.4     |      39.8      |       67.4       |
> |  40%   |    95.5     |     81.5     |      40.5      |       69.6       |
> |  50%   |    96.2     |     82.1     |      41.1      |       71.4       |
> |  60%   |    96.7     |     81.7     |      40.4      |       71.7       |
> |  70%   |    96.8     |     81.9     |      39.9      |       71.4       |
> |  80%   |    96.4     |     82.1     |      40.2      |       71.2       |
>
>
> ### Q3: The paper lacks of explanation on the generation process of partial label and instance-dependent partial label
>
> **A3:** In Appendix B of our paper, we have detailed three different generation methods for candidate label datasets. Below, we provide appropriate supplementary explanations.
>
> 1. **Uniform Sampling Strategy (USS):** For each sample, aside from the ground truth label, the remaining \( K-1 \) labels each have two possible states—either included in the candidate label set or not. Importantly, every possible candidate label set (regardless of its size) has an equal probability of being generated.
>
> 2. **Flip Probability Sampling Strategy (FPS):** For each instance, each false-positive label \( y \) is incorporated into the candidate label set with a fixed probability parameter η.
>
> 3. **Instance-Dependent Generation:** This method employs a lightweight neural network to generate instance-specific candidate label sets, which are tailored to the unique characteristics of each individual sample.
>
> We will include more detailed explanations of the generation process in the revised version.

---

> > ### Comment · Reviewer_icVt · 2025-08-04
> >
> > The authors have addressed all of my concerns and I will Keep my positive score.

---

> > > ### Author Response · Authors · 2025-08-05
> > > **Official Comment by Authors**
> > >
> > > We sincerely appreciate your thorough review and are grateful that our responses have addressed your concerns. Thank you for maintaining your positive score.

---

### Official Review · Reviewer_nyFD · 2025-06-02

**Rating:** 4
**Confidence:** 4

**Summary:**

The paper addresses Partial Label Learning (PLL), a weakly supervised learning framework where each training instance is associated with a set of candidate labels, only one of which is correct. The authors conduct comprehensive evaluations of 11 foundation models across 13 PLL approaches on 8 benchmark datasets under three PLL scenarios: standard PLL (ST-PLL), long-tailed PLL (LT-PLL), and instance-dependent PLL (ID-PLL). They introduce PartialCLIP, an efficient fine-tuning framework tailored for foundation models in PLL. The study reveals that foundation models significantly enhance PLL performance, exhibit similar effectiveness across different PLL methods, maintain stability across varying levels of label ambiguity, but are sensitive to the selection and adaptation of foundation models. Additionally, leveraging CLIP’s text-embedding capabilities for classifier initialization and candidate label filtering using zero-shot CLIP proves effective.

**Dataset Code Accessibility:**

Yes

**Ethical Considerations:**

No, there are no or only very minor ethics concerns

**Final Justification:**

The authors have addressed all of my concerns and I will Keep my positive score.

**Limitations Weaknesses:**

1.CLIP’s pre-training on broad categories limits its effectiveness on fine-grained datasets (e.g., FGVC Aircraft) due to overly specific class names (e.g., "747-300") that CLIP cannot contextualize.

2.Aggressive pruning (e.g., top-k selection) may remove true labels when CLIP’s zero-shot predictions are unreliable. Besides, it focuses on vision-language models (CLIP-like) but does not explore other foundation models (e.g., DINO). It would be better if the author provides more discussion of theoretical guarantees for PLL with foundation models.

3.Performance is highly tied to the pre-trained model’s architecture and training data. Poorly aligned models (e.g., ImageNet-pretrained ViT) may underperform compared to CLIP variants.

4.More partial label learning methods should be included for more comprehensive evaluation or discussion:
[1]Partial-Label Learning with Conformal Candidate Cleaning
[2]Exploiting the Potential Supervision Information of Clean Samples in Partial Label Learning
[3]Revisiting Sparsity Constraint Under High-Rank Property in Partial Multi-Label Learning

5.The candidate label filtering process using zero-shot CLIP may inadvertently remove ground-truth labels, especially when CLIP is less discriminative. This aggressive pruning can degrade performance if true labels are excluded. Besides, the effectiveness of label filtering is contingent on CLIP’s zero-shot classification performance, which may vary across different datasets and label complexities.

**Strengths Contributions:**

1.PartialCLIP is the first unified fine-tuning framework that systematically integrates vision-language foundation models (e.g., CLIP) into various PLL scenarios, including ST-PLL, LT-PLL, and ID-PLL.

2.The framework is loss-agnostic and model-agnostic, allowing compatibility with multiple partial label loss functions and different types of foundation models and fine-tuning methods.

3.The proposed method covers standard, long-tailed, and instance-dependent PLL settings, ensuring that the findings are relevant to a wide range of real-world applications.

---

> ### Author Rebuttal · Authors · 2025-07-29
>
> We are grateful for your professional comments, which significantly enhanced the paper’s quality and rigor. We’ve prepared below a recap of your viewpoints together with our relevant replies.
>
> ### Q1: CLIP’s pre-training on broad categories limits its effectiveness on fine-grained datasets (e.g., FGVC Aircraft) due to overly specific class names (e.g., "747-300") that CLIP cannot contextualize.
>
> **A1:** We have observed that the performance improvement of PartialCLIP on fine-grained datasets is indeed less significant than that on general datasets. The core reason for this phenomenon is that CLIP's pre-training is mainly based on a wide range of general categories, and what it possesses is general knowledge. However, it lacks corresponding professional domain knowledge and contextual understanding ability for overly specific class names (such as "747-300") in fine-grained datasets, which restricts its performance in such scenarios. This issue stems from a common limitation of CLIP. We aim to address this issue in future work.
>
>
> ### Q2&Q5: Aggressive pruning in zero-shot CLIP-based candidate label filtering may remove true labels when CLIP's predictions are unreliable, with its effectiveness tied to CLIP's performance (varying by dataset/label complexity); additionally, it focuses only on CLIP-like models without exploring others or providing theoretical guarantees for PLL with foundation models.
>
> **A2&A5:** As shown in the tables below, we report sensitivity experiments on the top-k accuracy of CLIP's zero-shot results and the corresponding performance of our model across the four datasets noted in Table 6 of the paper. Specifically, for each sample's CLIP zero-shot inference result, we examine the proportion of samples where the ground-truth label falls within the top k classes—with k ranging from 10% to 80% of the total number of classes K. This is precisely what "Top-k Accuracy" denotes in the title of the first table below. Experimental results show that when k = 50%*K, all four datasets ensure the ground-truth label is retained for at least 98.7% of samples. Additionally, we conduct experiments with DINO, and the results are presented in the following table. For the theoretical analysis of foundation models, we will address this in future work.
>
>
> Table: Top-k Accuracy of CLIP Zero-Shot Inference under Different k
>
> |        |   CIFAR10   |   CIFAR100   |    Places-LT   |    ImageNet-LT   |
> | :-----: | :---------: | :----------: | :------------: | :--------------: |
> |  10%   |    90.15    |     91.98    |      93.04     |       99.44      |
> |  20%   |    96.18    |     95.54    |      96.99     |       99.76      |
> |  30%   |    97.82    |     97.11    |      98.40     |       99.87      |
> |  40%   |    98.69    |     98.08    |      99.13     |       99.93      |
> |  50%   |    99.32    |     98.72    |      99.51     |       99.95      |
> |  60%   |    99.72    |     99.22    |      99.72     |       99.97      |
> |  70%   |    99.86    |     99.53    |      99.84     |       99.99      |
> |  80%   |    99.94    |     99.75    |      99.93     |       99.99      |
>
>
> Table: Model Performance under Different k
>
> |        |   CIFAR10   |   CIFAR100   |    Places-LT   |    ImageNet-LT   |
> | :-----: | :---------: | :----------: | :------------: | :--------------: |
> |  10%   |    92.2     |     80.2     |      34.3      |       71.0       |
> |  20%   |    93.7     |     81.3     |      39.3      |       68.7       |
> |  30%   |    94.6     |     81.4     |      39.8      |       67.4       |
> |  40%   |    95.5     |     81.5     |      40.5      |       69.6       |
> |  50%   |    96.2     |     82.1     |      41.1      |       71.4       |
> |  60%   |    96.7     |     81.7     |      40.4      |       71.7       |
> |  70%   |    96.8     |     81.9     |      39.9      |       71.4       |
> |  80%   |    96.4     |     82.1     |      40.2      |       71.2       |
>
> Table: Comparisons of Performance between DINO and OpenAI CLIP
>
>
> |                        | cifar10 (ST-PLL) | cifar100 (ST-PLL) | cub200 (ST-PLL) | fgvc100 (ST-PLL) | dogs120 (ST-PLL) | cifar100_lt (LT-PLL) | dogs120 (ID-PLL) |
> | :--------------------: | :--------------: | :---------------: | :-------------: | :--------------: | :--------------: | :------------------: | :--------------: |
> | OpenAI CLIP-ViT-B/16   |      97.3        |       88.7        |      85.1       |       79.9       |       84.4       |        74.8         |       78.9       |
> | DINO-ViT-B/16          |      96.2        |       73.5        |      42.1       |       37.6       |       66.6       |        38.6         |       61.6       |
>
> ### Q3: Performance is highly tied to the pre-trained model’s architecture and training data. Poorly aligned models (e.g., ImageNet-pretrained ViT) may underperform compared to CLIP variants.
>
> **A3:** We agree that the performance of a model is closely related to the backbone architecture, as well as the difficulty, size, and quantity of the dataset. In practice, we have observed that models like ResNet, ViT, and CLIP may exhibit comparable performance across different datasets—their relative effectiveness can vary depending on the specific characteristics of the task and data, rather than being universally superior or inferior. For example, Table 4 in the main text presents results for three versions of ViT, which, though slightly inferior overall to CLIP and its variants, still rank among the top on certain datasets such as DOGS120. This further highlights the importance of selecting a backbone that aligns well with the target dataset and task requirements.
>
> ### Q4: More partial label learning methods should be included for more comprehensive evaluation or discussion: [1]Partial-Label Learning with Conformal Candidate Cleaning [2]Exploiting the Potential Supervision Information of Clean Samples in Partial Label Learning [3]Revisiting Sparsity Constraint Under High-Rank Property in Partial Multi-Label Learning
>
> **A4:** We will integrate these three methods into our proposed PartialCLIP once their codes are all made open-source. In the final version of the paper, we will present the relevant results and conduct corresponding discussions.

---

### Official Review · Reviewer_p6Zs · 2025-06-22

**Rating:** 5
**Confidence:** 4

**Summary:**

This paper introduces PartialCLIP, a framework for Partial Label Learning (PLL) that leverages foundation models (e.g., CLIP) to address challenges in weakly supervised learning. The authors conduct extensive empirical evaluations across three PLL scenarios (standard, long-tailed, and instance-dependent PLL), comparing 13 PLL methods on 8 benchmark datasets using 11 foundation models. Key contributions include empirical benchmarking of foundation models in PLL, showing significant performance gains over traditional CNNs; PartialCLIP, a parameter-efficient fine-tuning framework that integrates vision-language alignment (e.g., text-embedding classifier initialization, candidate label filtering); and findings that PLL methods with foundation models achieve similar performance, are robust to label ambiguity, and depend heavily on model selection and adaptation strategies.

**Dataset Code Accessibility:**

Yes

**Ethical Considerations:**

No, there are no or only very minor ethics concerns

**Final Justification:**

I think the authors have addressed my concerns, I will keep my score.

**Limitations Weaknesses:**

1. The candidate label filtering strategy assumes CLIP’s zero-shot confidence scores are reliable, but this may fail for ambiguous or rare classes. A sensitivity analysis of the filtering threshold (e.g., how performance degrades if the top-k selection is too aggressive) would strengthen the approach.

2. GPU memory/time costs for larger models (e.g., MetaCLIP-ViT-B/16) are unreported.

3. Symbols like η (partial rate), γ (imbalance ratio) lack definitions when first used.

**Strengths Contributions:**

1. The work is significant as the first systematic study of foundation models in PLL, bridging a gap in weakly supervised learning. PartialCLIP’s integration of CLIP’s zero-shot capabilities (e.g., text-embedding initialization, label filtering) is innovative. The paper demonstrates that fine-tuning foundation models reduces the need for complex PLL algorithms, as their transferable representations dominate performance.

2. The empirical evaluation is rigorous, covering 3 PLL scenarios, 8 datasets, and 11 models (Tables 1–4). The authors highlight limitations of current PLL methods (e.g., sensitivity to foundation model choice, Fig. 1c) and propose solutions (e.g., label filtering, Table 6). The presentation is clear, with well-organized figures (Fig. 1, 2) and tables (Tables 1–6) that support key claims.

3. The work has practical impact, potentially reducing annotation costs in real-world applications (e.g., medical imaging, web mining). The open-source code enhances reproducibility.

4. The paper is well-written and easy to understand.

---

> ### Author Rebuttal · Authors · 2025-07-29
>
> Thank you for your valuable comments, which have greatly helped improve the manuscript. What follows is a condensation of your observations paired with our corresponding feedback.
>
> ### Q1: The candidate label filtering strategy’s assumption of reliable CLIP zero-shot confidence scores may fail for ambiguous/rare classes, and a sensitivity analysis of the filtering threshold would strengthen it.
>
> **A1:** Thank you for the suggestion! Below are tables presenting sensitivity experiments on the Top-k accuracy of CLIP's zero-shot results, alongside the corresponding performance of our model across the four datasets noted in Table 6 of the paper. Specifically, for each sample's CLIP zero-shot inference result, we analyze the proportion of samples where the ground-truth label is included among the top k classes—where k ranges from 10% to 80% of the total number of classes K. This is precisely what "Top-k Accuracy" refers to in the title of the first table below. Experimental results show that when k = 50%*K, all four datasets ensure that the ground-truth label is not excluded for at least 98.7% of samples.
>
>
> Table: Top-k Accuracy of CLIP Zero-Shot Inference under Different k
>
> |        |   CIFAR10   |   CIFAR100   |    Places-LT   |    ImageNet-LT   |
> | :-----: | :---------: | :----------: | :------------: | :--------------: |
> |  10%   |    90.15    |     91.98    |      93.04     |       99.44      |
> |  20%   |    96.18    |     95.54    |      96.99     |       99.76      |
> |  30%   |    97.82    |     97.11    |      98.40     |       99.87      |
> |  40%   |    98.69    |     98.08    |      99.13     |       99.93      |
> |  50%   |    99.32    |     98.72    |      99.51     |       99.95      |
> |  60%   |    99.72    |     99.22    |      99.72     |       99.97      |
> |  70%   |    99.86    |     99.53    |      99.84     |       99.99      |
> |  80%   |    99.94    |     99.75    |      99.93     |       99.99      |
>
>
> Table: Model Performance under Different k
>
> |        |   CIFAR10   |   CIFAR100   |    Places-LT   |    ImageNet-LT   |
> | :-----: | :---------: | :----------: | :------------: | :--------------: |
> |  10%   |    92.2     |     80.2     |      34.3      |       71.0       |
> |  20%   |    93.7     |     81.3     |      39.3      |       68.7       |
> |  30%   |    94.6     |     81.4     |      39.8      |       67.4       |
> |  40%   |    95.5     |     81.5     |      40.5      |       69.6       |
> |  50%   |    96.2     |     82.1     |      41.1      |       71.4       |
> |  60%   |    96.7     |     81.7     |      40.4      |       71.7       |
> |  70%   |    96.8     |     81.9     |      39.9      |       71.4       |
> |  80%   |    96.4     |     82.1     |      40.2      |       71.2       |
>
>
>
> ### Q2: GPU memory/time costs for larger models (e.g., MetaCLIP-ViT-B/16) are unreported
>
> **A2:** The GPU memory and time costs for different PLL baselines and backbones are presented in the tables below. Here, GPU memory denotes peak memory usage. While PartialCLIP incurs higher inference costs than PLL baselines, these costs remain within an acceptable range, with a significant improvement in performance (see Tables 1, 2, and 3 of the paper). For time costs, the per-batch time for training and testing stages is reported separately (with a batch size of 64). All experiments were conducted on a single Nvidia A6000, using the CIFAR100 dataset—with a training set of 50,000 samples and a test set of 10,000 samples.
>
>
> Table: Time and GPU Cost of Various PLL Baselines
>
> | Baselines | GPU (GB) | Train Time (s) | Test time (s) |
> | :-------: | :------: | :------------: | :-----------: |
> |    CC     |   8.4    |      0.51      |     0.20      |
> |    LWS    |   8.4    |      0.73      |     0.20      |
> |   CAVL    |   8.4    |      0.73      |     0.20      |
> |  CRDPLL   |  15.7    |      1.43      |     0.20      |
> |  PRODEN   |   5.1    |      0.34      |     0.28      |
> |  ABS-MAE  |   8.4    |      0.50      |     0.24      |
> |  ABS-GCE  |   8.4    |      0.50      |     0.33      |
>
>
> Table: Time and GPU Cost of Various Backbones
>
> | Pre-trained Model        | GPU (GB) | Train time (s) | Test time (s) |
> | :----------------------: | :------: | :------------: | :-----------: |
> | OpenAI CLIP-ViT-B16      |   8.4    |      1.43      |     0.20      |
> | OpenAI CLIP-ViT-B32      |   4.0    |      2.37      |     0.29      |
> | CLIP-ViT-B16-L400m       |   12.4   |      1.91      |     0.29      |
> | CLIP-ViT-B32-L400m       |   3.2    |      1.26      |     0.26      |
> | In21k-ViT-B16-augreg     |   12.1   |      1.93      |     0.30      |
> | In21k-ViT-T16-augreg     |   3.1    |      0.89      |     0.33      |
> | In21k-ViT-S16-augreg     |   6.0    |      1.61      |     0.27      |
> | MetaCLIP-ViT-B16         |   15.4   |      1.94      |     0.20      |
> | MetaCLIP-ViT-B32         |   3.8    |      1.80      |     0.21      |
> | SigLIP-ViT-B16           |   12.1   |      2.51      |     0.19      |
> | SigLIP2-ViT-B16          |   12.1   |      1.23      |     0.19      |
>
>
> ### Q3: Symbols like η (partial rate), γ (imbalance ratio) lack definitions when first used
>
> **A3:** Thank you for the suggestion! We will thoroughly check the manuscript and provide definitions for symbols such as η and γ when they are first mentioned in the main text.

---

### Official Review · Reviewer_Puz9 · 2025-07-03

**Rating:** 3
**Confidence:** 3

**Summary:**

This paper introduces PartialCLIP, a unified fine-tuning framework that leverages foundation models for partial label learning (PLL) across three scenarios: standard PLL (ST-PLL), long-tailed PLL (LT-PLL), and instance-dependent PLL (ID-PLL). The authors conduct an extensive empirical evaluation of 11 foundation models across 13 PLL approaches on 8 benchmark datasets. The framework incorporates parameter-efficient fine-tuning methods and proposes techniques for leveraging CLIP's vision-language alignment capabilities through text-based classifier initialization and candidate label filtering.

**Additional Feedback:**

- NIT: The legend on Figure 1c is 'microscopic'.
- Table 6: 100.9 should be 10.9, right?

**Dataset Code Accessibility:**

Yes

**Ethical Considerations:**

No, there are no or only very minor ethics concerns

**Final Justification:**

I appreciate the authors' extensive experiments conducted and presented in the main paper and during the rebuttal phase, I belive a lot of work was put into that submission. However, I believe that my main concerns about limited novelty (a relatively new application of pre-trained models) and the paper not entirely matching the datasets and the benchmark track remain. Because of that, I keep my score as it is.

**Limitations Weaknesses:**

- The novelty of PartialCLIP is a bit limited, at its PEFT + CLIP backbone, which shouldn't be a problem for a dataset or benchmark paper, but in this case,this method seems to be the most significant contribution, as the evaluation framework doesn't introduce any other novel elements.
- PartialCLIP is model agnostic, but the experiments are conducted only with CLIP. It would be nice to see at least one experiment with different backbones.
- While the authors claim that their approach is much cheaper to train, it's not clear what the cost of Inference, as inference times are not reported.
- a bit awkward *you need* type of title :)

I generally find it difficult to rate this paper; it feels to me like the authors are trying to publish their method, which is not very novel, without any deeper analysis in the paper disguised as a framework/benchmark paper.

**Strengths Contributions:**

- I really like that the framework implements some generation strategies of candidate label sets, especially Instance-Dependent Generation, as many papers in this area use mostly unrealistic uniform sampling strategies.
- The paper goes a bit beyond introducing just a framework. The evaluation study, I believe, goes a bit beyond what we have seen so far in related papers (however, I was not following this area recently), with PartialCLIP being a new algorithmic contribution.
- The code is well documented, seems to be well organized, easy to use, and the quick start actually works.

---

> ### Author Rebuttal · Authors · 2025-07-29
>
> We thank the reviewer for the insightful comments. Below, we provide a summary of your comments along with our corresponding responses.
>
> ### Q1: The evaluation framework doesn't introduce any other novel elements
>
> **A1:** Thank you for the comment. To the best of our knowledge, this is the first framework that systematically integrates vision-language models fine-tuning into PLL scenarios, breaking the limitation that PLL research has long relied on traditional CNN architectures. We argue that **the ability to learning from partial labels with pre-trained Transformers has not been thoroughly explored**. To fill this gap, we empirically perform comprehensive evaluations involving 11 foundation models, assessing their performance across 13 PLL approaches, 8 benchmark datasets, and 3 distinct PLL scenarios.
>
> Our key findings indicate that existing PLL approaches typically: 1) attain substantial performance improvements when integrated with foundation models; 2) display striking similarities in performance relative to one another; 3) retain stable performance across different levels of ambiguity; and 4) are highly influenced by the selection of foundation models and the adaptation strategies employed.
>
> The experimental results and corresponding analyses presented in this work highlight the constraints of current PLL approaches and offer valuable perspectives for the development of more generalizable PLL models. Hence, **we consider the proposed framework and conducted evaluations as promising contributions to the field of PLL.** For example, different PLL methods perform strikingly similarly on the base model (see Table 1 of the paper), indicating that the representational capability of the base model may be more important than the PLL algorithms themselves. This may lead to a shift in research focus from "how to design better PLL algorithms" to "how to better utilize foundation models to solve PLL problems."
>
> ### Q2: PartialCLIP is model-agnostic but tested only on CLIP; experiments with other backbones would help.
>
> **A2:** Thank you for your comment. In the paper, we use CLIP the default backbone for its good generalization performance.   However, the proposed learning framework can be readily applied to different backbones. For example, **in Table 4 of the paper, we report the results for imagenet21k-pretrained ViT, SigLIP, MetaCLIP, CLIP, and OpenAI CLIP**. Additionally, we conduct experiments with DINO, and the results are presented in the following table.
>
> Table: Comparisons of Performance between DINO and OpenAI CLIP
>
> |                        | cifar10 (ST-PLL) | cifar100 (ST-PLL) | cub200 (ST-PLL) | fgvc100 (ST-PLL) | dogs120 (ST-PLL) | cifar100_lt (LT-PLL) | dogs120 (ID-PLL) |
> | :--------------------: | :--------------: | :---------------: | :-------------: | :--------------: | :--------------: | :------------------: | :--------------: |
> | OpenAI CLIP-ViT-B/16   |      97.3        |       88.7        |      85.1       |       79.9       |       84.4       |        74.8         |       78.9       |
> | DINO-ViT-B/16          |      96.2        |       73.5        |      42.1       |       37.6       |       66.6       |        38.6         |       61.6       |
>
>
> ### Q3: While the authors claim that their approach is much cheaper to train, it's not clear what the cost of Inference, as inference times are not reported
>
> **A3:** Thank you for your comment. The GPU memory/time costs for different PLL baselines and different backbones are listed in the following tables, respectively. Herein, GPU memory refers to the peak memory. Although PartialCLIP requires more inference cost than PLL baselines, it is within an acceptable range, and the performance has been significantly improved (see Tables 1, 2, and 3 in the main text). For time costs, the time of each batch in the training and test stages is listed separately (with a batch size of 64). All experiments were run on one single Nvidia A6000, using the CIFAR100 dataset with a training set size of 50,000 and a test set size of 10,000. Notably, all baselines are implemented in our proposed PartialCLIP framework.
>
>
> Table: Time and GPU Cost of Various PLL Baselines
>
> | Baselines | GPU (GB) | Train Time (s) | Test time (s) |
> | :-------: | :------: | :------------: | :-----------: |
> |    CC     |   8.4    |      0.51      |     0.20      |
> |    LWS    |   8.4    |      0.73      |     0.20      |
> |   CAVL    |   8.4    |      0.73      |     0.20      |
> |  CRDPLL   |  15.7    |      1.43      |     0.20      |
> |  PRODEN   |   5.1    |      0.34      |     0.28      |
> |  ABS-MAE  |   8.4    |      0.50      |     0.24      |
> |  ABS-GCE  |   8.4    |      0.50      |     0.33      |
>
>
> Table: Time and GPU Cost of Various Backbones
>
> | Pre-trained Model        | GPU (GB) | Train time (s) | Test time (s) |
> | :----------------------: | :------: | :------------: | :-----------: |
> | OpenAI CLIP-ViT-B16      |   8.4    |      1.43      |     0.20      |
> | OpenAI CLIP-ViT-B32      |   4.0    |      2.37      |     0.29      |
> | CLIP-ViT-B16-L400m       |   12.4   |      1.91      |     0.29      |
> | CLIP-ViT-B32-L400m       |   3.2    |      1.26      |     0.26      |
> | In21k-ViT-B16-augreg     |   12.1   |      1.93      |     0.30      |
> | In21k-ViT-T16-augreg     |   3.1    |      0.89      |     0.33      |
> | In21k-ViT-S16-augreg     |   6.0    |      1.61      |     0.27      |
> | MetaCLIP-ViT-B16         |   15.4   |      1.94      |     0.20      |
> | MetaCLIP-ViT-B32         |   3.8    |      1.80      |     0.21      |
> | SigLIP-ViT-B16           |   12.1   |      2.51      |     0.19      |
> | SigLIP2-ViT-B16          |   12.1   |      1.23      |     0.19      |
>
>
> ### Q4: a bit awkward you need type of title
>
> **A4:** Thank you for your valuable comment. We will carefully consider your suggestion and revise the title in the next version.
>
> ### Q5: NIT: The legend on Figure 1c is 'microscopic'
>
> **A5:** Thank you for pointing out this flaw. We will correct the legend of Figure 1c in the next version.
>
> ### Q6: Table 6: 100.9 should be 10.9, right?
>
> **A6:** We have confirmed that it is indeed 100.9. Note that the ImageNet-LT dataset consists of 1000 classes. In the FPS strategy, except for the ground truth label, all false-positive labels have a 0.1 probability of being randomly flipped into the candidate label set, that is, (1000 - 1) * 0.1 = 99.9. Due to the premise of PLL, the ground truth label must be included, so the average size of the candidate label set is 99.9 + 1 = 100.9.

---

> > ### Author Response · Authors · 2025-08-07
> > **Looking Forward to Your Feedback**
> >
> > Dear Reviewer Puz9,
> >
> > Thank you for your thoughtful and detailed comments, which have been instrumental in improving the quality of our manuscript. We have carefully addressed all of your concerns. Specifically, we have clarified the novel elements of our framework, provided additional experimental results using different backbones, and included a discussion on inference cost. We have also carefully considered and responded to your feedback on the title, Figure 1, and Table 6.
> >
> > We sincerely appreciate your valuable feedback. Should any points still require clarification, we would be happy to provide further explanation.
> >
> > Best regards,
> >
> > The Authors

---

> > ### Comment · Reviewer_Puz9 · 2025-08-08
> >
> > Dear Authors, thank you for your response. I appreciate additional results and elaborations. I think you addressed my comments well, which only strengthens my opinion that a lot of work was put into this contribution. I don't have further comments. However, my main concern about limited novelty and the paper not entirely matching the datasets and the benchmark track remains, because of that, I keep my score as it is. However, I won't be opposing the paper being accepted, as it seems that other reviewers don't share these concerns.

---

> ### Comment · Area_Chair_WNvh · 2025-08-05
>
> Dear reviewer,
> please read the other reviews and the author response, and start a discussion with the authors promptly to allow time for an exchange.
> Your AC

---

> ### Author Response · Authors · 2025-08-09
>
> Dear Reviewer Puz9,
>
> Sincerely thank you for reviewing our paper and your detailed, insightful comments. Your feedback helped identify improvements and deepen our thinking. We’ve carefully considered your points, prepared detailed responses to clarify concerns about novelty and format alignment, and hope these aid your comprehensive understanding of our research.
>
> At the initial stage of review, you expressed difficulty in rating our manuscript and assigned a negative score. In our rebuttal, we addressed their concerns by clarifying the novel elements of our framework, providing additional experimental results using different backbones, and including a discussion on inference cost. We also responded to their specific feedback regarding the title, Figure 1, and Table 6. **Additionally, the title of the paper will be revised to make its expression more appropriate.** We are particularly grateful for your feedback, especially your comment that “you addressed my comments well”—this means a great deal to us and reinforces our commitment to refining the work based on constructive input.
>
> We’ve noted your concerns about insufficient novelty and dataset-benchmark alignment. Clarifications: **This paper pioneers a systematic comparative framework to evaluate Partial Label Learning (PLL) methods across foundation models and parameter-efficient fine-tuning strategies—aligning with benchmark paper norms.** Text-embedding classifier initialization and candidate label filtering strategies are not standalone innovations but CLIP-specific optimizations to boost PLL benchmark accuracy, not prioritize methods. This structure has precedents: prior NeurIPS benchmark papers advanced fields via rational evaluation systems and targeted optimizations. Our paper’s positioning and contributions align with such academic value.
>
> We are grateful that other reviewers have offered positive feedback on our work: they noted our attempt to pioneer the integration of vision-language foundation models into PLL across standard, long-tailed, and instance-dependent scenarios, recognizing it as an early comprehensive study in this domain (Reviewers p6Zs, nyFD, icVt). They mentioned the potential value of leveraging CLIP’s zero-shot capabilities—such as text-embedding initialization and label filtering—to reduce reliance on complex PLL algorithms (Reviewers p6Zs, Puz9), and observed that the loss/model-agnostic framework may enhance compatibility with various methods and models (Reviewer nyFD). **Additionally, they acknowledged the extensive empirical evaluations across 3 scenarios, 8 datasets, and 11 models as efforts to demonstrate robustness (Reviewers p6Zs, icVt).** We also appreciate their recognition of the manuscript’s clarity, organized figures/tables (Reviewers p6Zs, icVt), and open-source code aimed at aiding reproducibility and practical applications (Reviewers Puz9, p6Zs, icVt).
>
> In summary, your insightful feedback has significantly aided us in further refining the paper, and we sincerely appreciate the time and effort you have dedicated to reviewing and deliberating on our work. With regard to your concerns regarding the paper’s novelty and adherence to the required format, we deemed it necessary to provide corresponding clarifications, as detailed above. **Our greatest contribution should be proposing a framework based on pre-trained models and conducting extensive evaluations. While the methodology itself may lack substantial theoretical novelty, it addresses a critical underexplored research gap—no prior studies have attempted such an approach, and thus this work is expected to contribute to advancing the development of the field**. We hope these explanations will facilitate a deeper understanding of our paper and help alleviate any uncertainties you may have encountered during your initial evaluation. Thank you once again for your valuable input and constructive guidance.
>
> Best Regards,
> The Authors

---

### Note · Authors · 2025-08-14

We sincerely thank all reviewers, ACs, and SACs for their valuable time and careful evaluation of our work. This paper identifies a key shortfall in existing Partial Label Learning (PLL) research: prior studies have largely depended on conventional convolutional neural networks, while devoting limited focus to foundation models. To bridge this gap, we propose **PartialCLIP**, a systematic evaluative framework for assessing methods across foundation models and parameter-efficient fine-tuning strategies. We conduct comprehensive empirical evaluations of 11 foundation models across 13 PLL methods on 8 benchmark datasets under 3 distinct PLL scenarios.

We carefully addressed all feedback during the rebuttal and discussion phases. We are greatly encouraged by the reviewers' acknowledgment and appreciation of our comprehensive responses. Key responses to each reviewer's concerns are summarized below:

- **Reviewer Puz9**: We appreciate the comments on results from other foundation models, inference costs, title phrasing, novelty of the evaluation framework, and its benchmark conformity. We addressed each in detail, with thorough explanations in the second-round response highlighting our work's novelty and alignment with benchmark protocols.
- **Reviewer p6Zs**: We acknowledge requests for GPU memory/time costs, symbol clarifications, and sensitivity analysis of the top-k filtering threshold. In response, we conducted experiments and provided detailed results on computational requirements, definitions, and strategy robustness under varying thresholds.
- **Reviewer nyFD**: We value comments on using models like DINO, adding PLL baselines, explaining results, and top-k questions. We performed k-sensitivity analysis, DINO experiments to validate our framework, clarified pre-trained model characteristics and effects, and explained modest performance on fine-grained datasets.
- **Reviewer icVt**: We thank the reviewer for requesting k-value sensitivity analysis and partial-label generation clarification. We provided explanations and experiments verifying the strategy's feasibility, showing that at k = 50%*K (K = total classes), all four datasets retain ground-truth labels for at least 98.7% of samples.

Furthermore, detailed responses addressed other questions, including justifications for numerical values (e.g., 100.9), updates to the Figure 1(c) legend, and candidate label set generation methods. All suggestions will be integrated into the revised manuscript.

---

### Decision · Program_Chairs · 2025-09-18

**Decision:**

Reject

**Comment:**

This work introduces PartialCLIP, a framework for fine-tuning models with partial label learning (PLL). It also conducts an evaluation across 11 foundation models, 13 PLL approaches, and 8 benchmark datasets, showing that foundation models significantly improve performance but results are sensitive to model selection and adaptation strategies. Reviewers raised  several questions, e.g. regarding computational costs, evaluation protocols, and sensitivity analyses, but generally appreciated the responses. One reviewers found the novelty of the work limited and partially coherent with the scope of the DB track. Nevertheless, given the overall positive recommendations and the relevance of the topic, the AC recommends acceptance.

===== FINAL UPDATE FROM DB Track PCs ====

The final decision for this paper has been taken by the program chairs after consultation with the SACs. All Senior Area Chairs have ranked papers according to the feedback from the AC during the review process. We decided to leave the original meta-review to reflect the opinion of the AC in light of the initial discussions with reviewers and SAC.